



# Increasing the spatial resolution of cloud property retrievals from Meteosat SEVIRI by use of its high–resolution visible channel: Evaluation of candidate approaches with MODIS observations

Frank Werner[1,2] and Hartwig Deneke[1]

[1]Leibniz Institute for Tropospheric Research, Permoserstraße 15, 04318 Leipzig, Germany
[2]Now at Jet Propulsion Laboratory, 4800 Oak Grove Drive, Pasadena, CA 91109, USA

**Correspondence:** Frank Werner (frank.werner@jpl.nasa.gov)

**Abstract.** This study presents and evaluates several candidate approaches for downscaling observations from the Spinning Enhanced Visible and Infrared Imager (SEVIRI) in order to increase the horizontal resolution of subsequent cloud optical thickness ($\tau$) and effective droplet radius ($r_{\text{eff}}$) retrievals from the native $3 \times 3\,\text{km}^2$ spatial resolution of the narrowband channels to $1 \times 1\,\text{km}^2$. These methods make use of SEVIRI's coincident broadband high–resolution visible (HRV) channel. For four

example cloud fields, the reliability of each downscaling algorithm is evaluated by means of collocated $1 \times 1\,\text{km}^2$ MODIS radiances, which are re-projected to the horizontal grid of the HRV channel, and serve as reference for the evaluation. By using these radiances smoothed with the spatial response function of the native SEVIRI channels as retrieval input, the accuracy at the SEVIRI standard resolution can be evaluated and an objective comparison of the accuracy of the different downscaling algorithms can be made. For the example scenes considered in this study, it is shown that neglecting high-frequency variations below the SEVIRI standard resolution results in significant random absolute deviations of the retrieved $\tau$ and $r_{\text{eff}}$ of up to

$\approx 14$ and $\approx 6\,\mu\text{m}$, respectively, as well as biases. By error propagation, this also negatively impacts the reliability of the subsequent calculation of liquid water path ($W_{\text{L}}$) and cloud droplet number concentration ($N_{\text{D}}$), which exhibit deviations of up to $\approx 89\,\text{g}\,\text{m}^{-2}$ and $\approx 177\,\text{cm}^{-3}$, respectively. For $\tau$, these deviations can be almost completely mitigated by the use of the HRV channel as a physical constraint, and by applying most of the presented downscaling schemes. For the accuracy of $r_{\text{eff}}$,

the choice of downscaling scheme however is important: deviations are generally of similar magnitude or larger than those for retrievals at the SEVIRI standard resolution, indicative of their limited skill at predicting high–frequency spatial variability in $r_{\text{eff}}$. A strong degradation of accuracy of $r_{\text{eff}}$ is observed for some of the approaches, which also affects subsequent $W_{\text{L}}$ and $N_{\text{D}}$ estimates. As a result, an approach which constrains the $r_{\text{eff}}$ to the lower–resolution results is recommended. Overall, this study demonstrates that an increase in horizontal resolution of SEVIRI cloud property retrievals can be reliably achieved by

use of its HRV channel, yielding cloud properties which are preferable in terms of accuracy to those obtained from SEVIRI's standard-resolution. This work advances efforts to mitigate impacts of scale mismatches among channels of multi–resolution instruments on cloud retrievals.



# 1 Introduction

In studies of the role of clouds in the climate system, the bispectral solar reflective method described by Twomey and Seton (1980); Nakajima and King (1990); Nakajima et al. (1991) is widely used to infer cloud optical and physical properties from satellite–based sensors. Based on observations of solar reflectance ($r$) from a channel pair at wavelengths with conservative

scattering (usually around $0.6\,\mu\mathrm{m}$ or $0.8\,\mu\mathrm{m}$) and significant absorption by cloud droplets (common channels are $1.6\,\mu\mathrm{m}$, $2.2\,\mu\mathrm{m}$, and $3.7\,\mu\mathrm{m}$), respectively, this method simultaneously estimates the cloud optical depth ($\tau$) and effective droplet radius ($r_{\mathrm{eff}}$) of a sampled cloudy pixel. This method however relies on a number of assumptions which are often violated in nature: clouds are considered to be horizontally homogeneous and to have a prescribed vertical structure, which is generally assumed to be vertically homogeneous or to show a linear increase of liquid water content as predicted by adiabatic theory

(see the discussions in Brenguier et al., 2000; Miller et al., 2016). Moreover, the observed cloud top reflectance field is usually described by one–dimensional (1D) plane–parallel radiative transfer, which neglects horizontal photon transport between neighboring atmospheric columns.

Use of the independent pixel approximation (IPA, see Cahalan et al., 1994a, b) produces uncertainties in the retrieved cloud variables that are dependent upon the horizontal resolution of the observing sensor. For sensors with a high spatial resolution,

the observations resolve the actual cloud heterogeneity, which are unaccounted for in the IPA approach. This usually results in an overestimation of both $\tau$ and $r_{\mathrm{eff}}$, as reported in Barker and Liu (1995); Chambers et al. (1997); Marshak et al. (2006). Conversely, for observations with a low spatial resolution, the actual cloud heterogeneity cannot be resolved. As a result, an underestimation (overestimation) of retrieved $\tau$ ($r_{\mathrm{eff}}$) is usually observed (Marshak et al., 2006; Zhang and Platnick, 2011; Zhang et al., 2012; Werner et al., 2018b). The analysis in Varnai and Marshak (2001) suggests that a horizontal scale of around

$1-2\,\mathrm{km}$ minimizes the combined uncertainty from unresolved and resolved cloud heterogeneity. While strategies to mitigate the effects of unresolved cloud variability have been recently reported in Zhang et al. (2016); Werner et al. (2018a), these techniques become less successful with lower–resolution sensors like those operated on geostationary satellites.

Remote sensing from geostationary platforms such as the Meteosat Spinning Enhanced Visible and Infrared Imager (SE-VIRI) offers unique capabilities for cloud studies not available from polar orbiting satellites. These advantages include more

frequent temporal sampling of individual regions and the ability to capture the temporal evolution (Bley et al., 2016; Senf and Deneke, 2017) and diurnal cycle of cloud parameters (Stengel et al., 2014; Bley et al., 2016; Martins et al., 2016; Seethala et al., 2018). However, SEVIRI pixels are characterized by a lower spatial resolution of its narrow–band channels compared to other operational remote sensing instrumentation, like the Moderate Resolution Imaging Spectroradiometer (MODIS, Platnick et al., 2003) or the Visible Infrared Imaging Radiometer Suite (VIIRS, Lee et al., 2006). Given the increase in retrieval uncertainty

due to the IPA constraints, there is a desire to increase the resolution for geostationary cloud observations.

The aim of this manuscript is to critically evaluate several candidate approaches for downscaling of the SEVIRI narrow–band reflectances for operational usage and to identify the most promising of these schemes, exploiting the fact that information on small-scale variability is available from its broadband high–resolution visible (HRV) channel. Of main concern is the ability to accurately capture information on the small–scale reflectance variability in the $1.6\,\mu\mathrm{m}$–channel, which predominantly arises





from variations in effective droplet radius. Conversely, cloud optical depth is expected to be well–constrained by the HRV channel, as it can be modelled by a linear combination of the $0.6\,\mu m$ and $0.8\,\mu m$ channels with good accuracy (Cros et al., 2006). This situation is similar to that found with other satellite instruments featuring multiple resolutions for the conservative and absorbing channels, such as the MODIS instrument (with $250\,m$ resolution versus $500\,m$ for $1.6\,\mu m$ or $1\,km$ for $2.1\,\mu m$),

VIIRS ($375\,m$ versus $750\,m$), and GOES–R ($500\,m$ versus $1\,km$). Therefore, we believe that our findings are also relevant there. This work is a companion paper to Deneke et al. (2019), which describes the overall retrieval scheme for obtaining cloud properties and solar radiative fluxes from the Meteosat SEVIRI instrument at the spatial resolution of its HRV channel, which will be established based on the findings of this study. The companion paper also presents an important extension of this approach to the retrieval of solar surface irradiance, based on the schemes presented in Deneke et al. (2008) and Greuell et al.

(2013). Satellite products with high temporal and spatial resolution are of particular interest for forecasting the production of solar power.

     A critical requirement, formulated at the start of this work, is to maintain a target accuracy for the retrieved effective radius based on the lower–resolution observations, while hoping for further improvements. This goal was set because the error in effective radius will propagate into other cloud products such as vertically integrated liquid or ice water path or the cloud droplet

number concentration, thereby potentially corrupting any gains in accuracy obtained from the improved spatial resolution. However, without an independent reference data set, it is impossible to determine whether this target can be met. Thus, higher–resolution reflectance observations from Terra–MODIS are remapped to SEVIRI's HRV and standard resolution grids here as basis for a thorough evaluation of the accuracy of the retrieved cloud parameters. This allows us to objectively benchmark the accuracy of candidate approaches by comparison of results from a true $1\,km$ resolution reflectance data set, and processed with

an identical retrieval scheme.

     The structure of the paper is as follows: section 2 describes both the SEVIRI and MODIS instruments used as basis for this study, as well as the covered observational domain. A brief overview of the SEVIRI cloud property retrieval algorithm is given in section 3, followed by a description of the different candidate approaches for the downscaling of the narrow–band SEVIRI channel observations in section 4. An example of lower– and higher–resolution cloud property retrievals is presented in section

5. Finally, a statistical evaluation of the different downscaling approaches based on remapped MODIS observations is given in section 6 for a limited number of example cloud fields. The manuscript presents the main conclusions and an outlook in section 7.

## 2   Data

This section gives an overview of both the SEVIRI and MODIS instruments in section 2.1 and 2.2. Here, the respective spectral

channels of interest for this study are listed. Subsequently, the observational domain is described in section 2.3.





## 2.1 SEVIRI

The current version of European geostationary satellites is the Meteosat Second Generation, which has provided operational data since 2004 (Schmetz et al., 2002). The SEVIRI imager is installed aboard the Meteosat–8 to Meteosat–11 platforms, which are positioned above longitudes of $9.5°$E and $0.0°$ longitude, respectively. One SEVIRI instrument samples the full disk

of the Earth from $0.0°$ longitude with a temporal resolution of fifteen minutes. However, a backup satellite positioned at $9.6°$E also scans a Northern subregion with a temporal resolution of five minutes (the so–called Rapid Scan Service). These samples – in our case from Meteosat–9 – provide the observational SEVIRI data set for the following analysis.

This study mainly considers observations from SEVIRI's solar reflectance channels 1–3, as well as from the spectrally broader HRV band. These channels cover the visible to near-infrared (VNIR) and shortwave-infrared (SWIR) spectral wave-

length ranges. The two VNIR reflectances ($r_{06}$ and $r_{08}$) are sampled in bands 1 and 2, respectively, and are centered around wavelengths $\lambda = 0.635\,\mu m$ and $\lambda = 0.810\,\mu m$. SWIR reflectances ($r_{16}$) are provided by channel 3 observations, which are centered around $\lambda = 1.640\,\mu m$. The horizontal resolution of the channel 1–3 samples is $3 \times 3\,km^2$. Conversely, the broadband reflectances $r_{HV}$ are sampled at SEVIRI's HRV channel at a horizontal scale of $1 \times 1\,km^2$. These observations cover the spectral range of $0.4 - 1.1\,\mu m$. Further information about the spectral width of each channel and the respective spectral and spatial

response functions can be found in Deneke and Roebeling (2010).

## 2.2 Terra–MODIS

The 36–band scanning spectroradiometer MODIS, which was launched aboard NASA's Earth Observing System satellites Terra and Aqua, has a viewing swath width of $2,330\,km$, yielding global coverage every two days. MODIS collects data in the spectral region between $0.415 - 14.235\,\mu m$, covering the VNIR to thermal–infrared spectral wavelength range. In general,

the spatial resolution at nadir of a MODIS pixel is $1,000\,m$ for most channels, although the pixel dimensions increase towards the edges of a MODIS granule. Only observations from the Terra satellite launched in 1999 are used here, due to broken detectors of the $1.64\,\mu m$ channel of the MODIS instrument on the Aqua satellite. Information on MODIS and its cloud product algorithms is given in (Ardanuy et al., 1992; Barnes et al., 1998; Platnick et al., 2003). The current version of the level 1b radiance and level 2 cloud products used is Data Collection 6.1 (C6.1).

## 25 2.3 Domain

In this study, data from a subregion of the full SEVIRI disk has been selected. This region, which is located within the European subregion described in Deneke and Roebeling (2010), is illustrated by the red borders in Figure 1. It is centered around Germany due to its intended domain of application (thus, from here on it is referred to as Germany domain) and comprises the latitude and longitude ranges of $\approx 44.30 - 57.77°$ and $\approx -0.33 - 21.65°$, respectively. This domain includes

$240 \times 400$ lower–resolution pixels (i.e., samples at a horizontal resolution of $3 \times 3\,km^2$) and is far away from the edges of the full SEVIRI disk, ensuring that the observed viewing zenith angles are $< 70°$. A relatively small domain was chosen, because the number of pixels to be processed will expand by a factor of $3 \times 3$, increasing the computational costs of the subsequent





cloud property retrievals by roughly one order of magnitude. Except for some regional dependencies introduced by changes in the prevalence of specific cloud types, we expect results of our study to also be valid for other domains.

## 3    SEVIRI cloud property retrieval algorithm

Retrieved cloud variables in this study are provided by the Cloud Physical Properties retrieval algorithm (CPP; Roebeling et al.,
2006), which is developed and maintained at the Royal Dutch Meteorological Institute (KNMI). It is used as basis for the CLAAS–1 and CLAAS–2 climate data records (Stengel et al., 2014; Benas et al., 2017) distributed by the Satellite Application Facility on Climate Monitoring (Schulz et al., 2009). Using a lookup table (LUT) of reflectances simulated by the Doubling–Adding KNMI (DAK: Smith and Timofeyev, 2001) radiative transfer model, observed and simulated reflectances at $0.6\,\mu\mathrm{m}$ and $1.6\,\mu\mathrm{m}$ are iteratively matched to yield estimates of $\tau$ and $r_{\mathrm{eff}}$. The CPP retrieval uses the cloud mask and cloud top
height products obtained from the software package developed and distributed by the satellite application facility of Support to Nowcasting and Very Short Range Forecasting (NWCSAF), Version 2016, as input (Le Gléau, 2016). The former product identifies cloudy pixels for the retrieval, while the information on the height of the cloud is used to account for the effects of gas absorption in the SEVIRI channels. An improved cloud detection scheme for the resulting higher–resolution SEVIRI retrievals based on the HRV channel based on Bley and Deneke (2013), with modifications described in Deneke et al. (2019)
(i.e., the companion paper that describes the final retrieval algorithm), has been integrated into the retrieval, but has not been used for this study.

For obtaining the results presented in this study, an experimental version of the retrieval that was developed in a separate branch has been used. This algorithm deviates in some aspects from the setup described in the companion paper. Specifically, it uses the default climatology of ancillary data sets available as part of the CPP retrieval system, which have a lower horizontal
resolution and do not match the specific time of the retrieval. This is expected to have only minor influence on the results presented here, because the absolute accuracy of the retrieval is not the primary focus of this study.

## 4    Candidate methods for downscaling SEVIRI reflectances

This section describes the necessary steps to convert the reflectances $r_{06}$, $r_{08}$, and $r_{16}$, available at the native SEVIRI resolution of $3 \times 3\,\mathrm{km}^2$, to reliable estimates of higher–resolution reflectances $\hat{r}_{06}$, $\hat{r}_{08}$, and $\hat{r}_{16}$, together with matching cloud properties,
at the spatial scale of $1 \times 1\,\mathrm{km}^2$ of the HRV channel. This downscaling process utilizes the high–resolution $r_{\mathrm{HV}}$ observations.

As a first step, all reflectances are interpolated to the HRV grid using trigonometric interpolation, implemented based on the discrete Fourier transform (see Deneke and Roebeling, 2010, for details). While this step increases the spatial sampling resolution, it does not add any additional high–frequency variability. In fact, after interpolation, the reflectance values of the central pixel of each $3 \times 3$ pixel block equal those of the corresponding standard–resolution pixel reflectances. However, the
pixels apart from the central one contain information about the large–scale reflectance variabilty and can be considered as a





baseline high–resolution approach. This approach already improves the agreement with true higher–resolution retrievals, as will be shown later in this study.

Three conceptually different downscaling techniques to improve upon this baseline method are described: (i) a statistical downscaling approach based on globally determined covariances between the SEVIRI reflectances in section 4.1, (ii) a local
method based on assumptions about the ratio of reflectances at different scales in section 4.2, and (iii) a technique combining globally determined covariances between the VNIR reflectances and the shape of the SEVIRI LUT, while assuming a constant $r_{\mathrm{eff}}$ within a standard SEVIRI pixel in order to constrain the SWIR reflectance in section 4.3. As variations of this last technique, two additional approaches are considered to improve upon the constant $r_{\mathrm{eff}}$ constraint in section 4.4. As will be shown, each of these approaches has advantages and disadvantages, and the impact on the cloud property retrievals will be evaluated in section
6 for a number of example scenes by means of collocated MODIS observations.

The derived reflectances $\hat{r}_{06}$ and $\hat{r}_{08}$, as well as $\hat{r}_{16}$, include an estimate of the spectrally dependent, high–frequency variability of an image, and are based on the actually observed $r_{\mathrm{HV}}$. These reflectances are different from those obtained by trigonometric interpolation of the respective channel observations at the native scale to the horizontal resolution of the HRV channel (i.e., the baseline approach), which are denoted by $\tilde{r}_{06}$, $\tilde{r}_{08}$, and $\tilde{r}_{16}$. While these variables also have a horizontal
resolution of $1 \times 1\,\mathrm{km}^2$, they only capture the low–frequency variability resolved by the SEVIRI sensor.

## 4.1 Statistical downscaling

The statistical downscaling algorithm for the two SEVIRI VNIR reflectances was first reported in Deneke and Roebeling (2010) and assumes a least-squares linear model that links $r_{06}$ and $r_{08}$ to the reflectances in the HRV channel (see Cros et al., 2006) in the form:

$$\langle \tilde{r}_{\mathrm{HV}} \rangle = a \cdot r_{06} + b \cdot r_{08}. \qquad (1)$$

Here, the HRV channel observations are first filtered with the spatial response function of the lower–resolution channels, which yields reflectances $\tilde{r}_{\mathrm{HV}}$ at the same $1 \times 1\,\mathrm{km}^2$ horizontal resolution, adjusted to the low–frequency variability at the spatial scale of the channel 1–3 observations. Subsampling the central pixel of each $3 \times 3 = 9$ pixel block subsequently yields $\langle \tilde{r}_{\mathrm{HV}} \rangle$ at the same $3 \times 3\,\mathrm{km}^2$ horizontal resolution as $r_{06}$ and $r_{08}$ (here, the subsampling of the field is denoted by $\langle \rangle$). The variables $a$
and $b$ are fit coefficients that are determined empirically by a least–squares linear fit. In order to derive a statistically significant and stable linear model, the coefficients $a$ and $b$ are calculated hourly between $08:00 - 16:00\,\mathrm{UTC}$ within 16–day intervals. Results for the time step $08:00\,\mathrm{UTC}$ are derived from 5–minute SEVIRI rapid–scan data between $08:00 - 08:25\,\mathrm{UTC}$, while the $16:00\,\mathrm{UTC}$ time step is comprised of SEVIRI observations between $15:30 - 16:00\,\mathrm{UTC}$. For all time steps in between, data is from all samples after minute 25 of the prior hour up to minute 25 of the current hour (e.g., fit coefficients for time step
$09:00\,\mathrm{UTC}$ are calculated from SEVIRI observations between $08:30 - 09:25\,\mathrm{UTC}$).

Values of hourly–derived fit coefficients for the Germany domain between 1 April and 31 July 2013 are shown in Figure 2(a) and 2(b) for $a$ and $b$, respectively. Here, circles represent the respective fit coefficient for each 16–day interval, which is indicated by the first Julian day in the time period. Colors highlight the different UTC time steps. It is obvious that both





coefficients $a$ and $b$ are very stable and show no noticeable variation from hour to hour, as well as from one 16–day interval to another. The median fit coefficients are $0.63$ (for $a$) and $0.40$ (for $b$), with low interquartile ranges (IQR) of $0.03$. The only exceptions are the fit coefficients derived for the first time period of 1–17 April 2013, especially for the morning and afternoon hours of $08:00 - 09:00$ and $15:00 - 16:00$ UTC. Here, $a$ and $b$ deviate significantly from the other results, with values of

$\approx 0.50$ and $\approx 0.52$, respectively, likely due to an abundance of observations with a large solar zenith angles of $\theta_0 > 60°$ in the eastern part of the domain.

The high–frequency reflectance variations for the SEVIRI HRV channel ($\delta r_{\mathrm{HV}}$) are calculated as the difference between the observed $r_{\mathrm{HV}}$ and $\tilde{r}_{\mathrm{HV}}$, which only resolves the low–frequency variability:

$$\delta r_{\mathrm{HV}} = r_{\mathrm{HV}} - \tilde{r}_{\mathrm{HV}}. \tag{2}$$

Following the linear model in Eq.(1), the high–frequency variations of the channel 1 and 2 reflectances ($\delta r_{06}$ and $\delta r_{08}$) are linked to $\delta r_{\mathrm{HV}}$ via:

$$
\begin{aligned}
\delta r_{06} &= S_{06} \cdot \delta r_{\mathrm{HV}} \\
\delta r_{08} &= S_{08} \cdot \delta r_{\mathrm{HV}}.
\end{aligned}
\tag{3}
$$

The optimal slopes $S_{06}$ and $S_{08}$, which minimize the least–squares deviations, can be derived from bivariate statistics:

$$
\begin{aligned}
\quad k_1 &= \sqrt{\frac{b^2 \cdot \mathrm{var}(r_{08})}{a^2 \cdot \mathrm{var}(r_{06})}} \\[2mm]
S_{06} &= \frac{1 + k_1 \cdot \mathrm{cor}(r_{06}, r_{08})}{a \cdot \left[1 + k_1{}^2 + 2 k_1 \cdot \mathrm{cor}(r_{06}, r_{08})\right]} \\[2mm]
k_2 &= \sqrt{\frac{a^2 \cdot \mathrm{var}(r_{06})}{b^2 \cdot \mathrm{var}(r_{08})}} \\[2mm]
S_{08} &= \frac{1 + k_2 \cdot \mathrm{cor}(r_{08}, r_{06})}{b \cdot \left[1 + k_2{}^2 + 2 k_2 \cdot \mathrm{cor}(r_{08}, r_{06})\right]}.
\end{aligned}
\tag{4}
$$

Here, $\mathrm{cor}(r_{06}, r_{08})$ is the linear correlation coefficient between the channel 1 and 2 reflectances, while $\mathrm{var}(r_{06})$ and $\mathrm{var}(r_{08})$

are the spatial variances of the respective samples. Note, that the sampling resolution of all reflectances is $3 \times 3\,\mathrm{km}^2$.

As a result, the high–resolution reflectances $\hat{r}_{06}$ and $\hat{r}_{08}$, which include the high–frequency variations, can be derived from the interpolated reflectances as:

$$
\begin{aligned}
\hat{r}_{06} &= \tilde{r}_{06} + \delta r_{06} \\
\hat{r}_{08} &= \tilde{r}_{08} + \delta r_{08}.
\end{aligned}
\tag{5}
$$

Note, that only $\hat{r}_{06}$ is used for the retrieval.

Similar steps can be applied for the calculation of $\hat{r}_{16}$. Again, a simple linear model is assumed to connect $r_{16}$ to the lower–resolution $\langle \tilde{r}_{\mathrm{HV}} \rangle$ at the spatial scales of the channel 1–3 observations:

$$\langle \tilde{r}_{\mathrm{HV}} \rangle = c \cdot r_{16}. \tag{6}$$

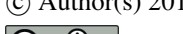



The symbol $c$ is used to denote the respective fit coefficient, which needs to be determined empirically. Similar to the coefficients $a$ and $b$ from the linear model for the VNIR reflectances, $c$ is calculated hourly between $08:00-16:00$ UTC within 16–day intervals. It has to be noted, however, that in contrast to the VNIR reflectances, this fit does not have a clear physical motivation, as there is no spectral overlap with the HRV channel.

The temporal behavior of the fit coefficient $c$ for the Germany domain for the time period between 1 April and 31 July 2013 is shown in Figure 2(c). In contrast to the coefficients $a$ and $b$, there is a noticeable trend in the data, both diurnally and during the transition from 1 April to 31 July. Diurnally, the variability in the hourly derived $c$ values ranges between $\mathrm{IQR}=0.05-0.15$, while the median 16–day value varies between $1.04$ and $1.25$. Overall, the median $c$ is $1.16$, with an IQR of $0.08$ (i.e., almost three times larger than the one for the coefficients $a$ and $b$). The observed trends and larger IQR in the $c$ data set shown in

Figure 2(c) illustrate that the linear model in Eq.(6) is not ideal, and is expected to introduce significant uncertainties in the calculation of $\tilde{r}_{16}$.

Values of $\tilde{r}_{16}$ can be derived similarly to Eqs.(3–5) for the channel 1 and 2 observations:

$$
\begin{aligned}
\delta r_{16} &= S_{16} \cdot \delta r_{\mathrm{HV}} \\
S_{16} &= \frac{\mathrm{cov}(r_{16}, \langle \tilde{r}_{\mathrm{HV}} \rangle)}{\mathrm{var}(r_{16})} \\
\hat{r}_{16} &= \tilde{r}_{16} + \delta r_{16}.
\end{aligned}
\tag{7}
$$

Note, that the use of linear models and bivariate statistics means that the downscaling algorithm described in this section is an example of statistical downscaling techniques, which are common in climate science applications (e.g., Benestad, 2011). While for the VNIR channels the spectral overlap with the HRV channel and the spectrally flat properties of clouds provide a sound physical justification for this technique, this is not the case for the SWIR channel.

The reliability of the linear model in Eq.(1) depends upon the correlation between channel 1 and 2 reflectances (i.e., $\mathrm{cor}(r_{06}, r_{08})$), as well as the stability of the fit coefficients $a$ and $b$. The analysis in Deneke and Roebeling (2010) concludes that the explained variance in the estimates of $\hat{r}_{06}$ and $\hat{r}_{08}$ are close to $1$, corresponding to low residual variances, which indicates that the linear model is robust. Moreover, the two fit coefficients are found to exhibit very low variability, as shown in Figures 2(a)–(b).

To verify the reliability of the linear model with a large SEVIRI data set, joint PDFs of the actually observed $\langle \tilde{r}_{\mathrm{HV}} \rangle$ and the results from Eq.(1) are shown in Figures 3(a)–(b); data is from all SEVIRI observations within the Germany domain during June 2013. In case of an ideal linear model, as well as a perfect correlation between the two reflectances, Eq.(1) would replicate the $\langle \tilde{r}_{\mathrm{HV}} \rangle$ observations. Conversely, deviations from these assumptions will yield different results from the sampled SEVIRI reflectances. It is clear that the linear model can reliably reproduce $\langle \tilde{r}_{\mathrm{HV}} \rangle$, as most of the observations lie on the 1:1 line,

and Pearson's product–moment correlation coefficient ($R$) is $R = 0.999$. While some larger deviations exist, such occurrences are significantly less likely (i.e., the joint probability density is several orders of magnitude lower than the most–frequent occurrences along the 1:1 line). Regarding $r_{16}$, the assumption of a linear model is evidently flawed, because the relationship between VNIR and SWIR reflectances depends on the optical and microphysical cloud properties. As a result, a single linear slope, which describes the whole relationship between the two reflectances for all cloud properties, will introduce significant





uncertainties. This is illustrated in Figure 3(c), where the Joint PDF of $\langle \tilde{r}_{\mathrm{HV}} \rangle$ and the results from the linear model in Eq.(6) are shown. The comparison between the two data sets reveals a much larger spread around the 1:1 line and a lower correlation coefficient. Overall, the relationship resembles the shape of a LUT, displayed in form of the well–known diagram introduced by Nakajima and King (1990), where changes in $r_{\mathrm{eff}}$ result in a spread in the observed SWIR reflectances (see, e.g., Werner et al.,

5    2016).

    To test the impact of changes in $a$ and $b$ on the derived $\hat{r}_{06}$ and $\hat{r}_{08}$, two experiments are conducted: (i) the fit coefficients are derived only from cloudy pixels and are compared to the higher–resolution results from $a$ and $b$, which are derived for all pixels. (ii) the Germany domain is divided into $100 \times 100\,\mathrm{km}^2$–subscenes and the fit coefficients are derived more locally within each subscene instead of globally from the full domain. Subsequently, statistics from the difference between the two data sets

are calculated. Data is from 14 June 2013 at 14:05 UTC. For experiment (i), the $1^{\mathrm{st}}$, $50^{\mathrm{th}}$, and $99^{\mathrm{th}}$ percentiles of the relative difference in $\hat{r}_{06}$ (defined as the difference between the reflectances from only cloudy data and the full data set, normalized by the full data set) are $-0.08, -0.02, 0.03\%$, while for $\hat{r}_{08}$ the analysis yields $-0.04, 0.02, 0.19\%$. Similarly, experiment (ii) yields relative differences of $-0.08, 0.03, 0.36\%$ and $-0.17, 0.00, 0.19\%$ for $\hat{r}_{06}$ and $\hat{r}_{08}$, respectively. These deviations are negligible compared to the measurement uncertainty and naturally, the correlation coefficients between the different data sets

are $R \approx 1.00$. This confirms the robustness of the linear model described in Eq.(1). For the derivation of $\hat{r}_{16}$ from Eq.(6), a slightly increased sensitivity to the fit coefficient $c$ is observed. Here, experiment (i) yields percentiles of the relative difference of $-0.16, 0.08, 0.86\%$, whereas experiment (ii) results in $-0.39, -0.01, 0.40\%$. While slightly higher deviations are observed compared to the linear model for the VNIR reflectances, the uncertainty in $\hat{r}_{16}$ induced by the variability in $c$ is still significantly lower than the measurement uncertainty.

## 4.2    Constant Reflectance Ratio Approach

Compared to the downscaling approach in section 4.1, where fit coefficients for a linear model are derived over a large temporal and spatial domain, this second method uses local relationships (i.e., on the pixel level) between the SEVIRI reflectances. The *Constant Reflectance Ratio Approach* was introduced by Werner et al. (2018b) and is based on the assumption that the inhomogeneity index of the HRV reflectance ($H_{\sigma,\mathrm{HV}}$, defined as the ratio of standard deviation $\sigma_{\mathrm{HV}}$ to the average, pixel–level

reflectance $\langle \tilde{r}_{\mathrm{HV}} \rangle$) equals that for the channel 1 reflectance ($H_{\sigma,06}$). This implies a spectrally consistent subpixel reflectance variability. The relationship can be written as:

$$
\begin{aligned}
H_{\sigma,06} &= H_{\sigma,\mathrm{HV}} \\
\frac{\sigma_{06}}{r_{06}} &= \frac{\sigma_{\mathrm{HV}}}{\langle \tilde{r}_{\mathrm{HV}} \rangle} \\
\frac{\sqrt{\frac{1}{9-1} \cdot \sum_{i=1}^{i=9} (\hat{r}_{06,i} - r_{06})^2}}{r_{06}} &= \frac{\sqrt{\frac{1}{9-1} \cdot \sum_{i=1}^{i=9} (r_{\mathrm{HV},i} - \langle \tilde{r}_{\mathrm{HV}} \rangle)^2}}{\langle \tilde{r}_{\mathrm{HV}} \rangle},
\end{aligned}
\tag{8}
$$

where the index $i = 1, 2, \ldots, 9$ indicates any one of the nine available $1 \times 1\,\mathrm{km}^2$–subpixels within a lower–resolution SEVIRI pixel (i.e., at a scale of $3 \times 3\,\mathrm{km}^2$). This relationship can be further simplified, assuming that this relationship is also true for





individual pixels:

$$\frac{\hat{r}_{06,i} - r_{06}}{r_{06}} = \frac{r_{\mathrm{HV},i} - \langle \tilde{r}_{\mathrm{HV}} \rangle}{\langle \tilde{r}_{\mathrm{HV}} \rangle}$$

$$\frac{\hat{r}_{06,i}}{r_{\mathrm{HV},i}} = \frac{r_{06}}{\langle \tilde{r}_{\mathrm{HV}} \rangle}. \tag{9}$$

The relationship in Eq.(9) suggests that the ratio of channel 1 and HRV reflectances (i.e., narrowband and broadband VNIR

reflectances) remains constant for different scales. Thus, this approach is called the *Constant Reflectance Ratio Approach*.

Finally, we can mitigate some of the scale effects by substituting the lower–resolution variables with the higher–resolution reflectances that resolve the low–frequency variability (i.e., $\tilde{r}_{06}$ and $\tilde{r}_{\mathrm{HV}}$) and solve for $\hat{r}_{06}$:

$$\hat{r}_{06} = r_{\mathrm{HV}} \cdot \frac{\tilde{r}_{06}}{\tilde{r}_{\mathrm{HV}}}. \tag{10}$$

Similarly, higher–resolution SWIR reflectances $\hat{r}_{16}$ can be derived from:

$$\hat{r}_{16} = r_{\mathrm{HV}} \cdot \frac{\tilde{r}_{16}}{\tilde{r}_{\mathrm{HV}}}. \tag{11}$$

As before, the relationship implies that the ratio of VNIR and SWIR reflectances remains constant for different scales. This assumption has been shown to be reasonable, at least for liquid water clouds over the ocean (Werner et al., 2018b).

A comparison of $\hat{r}_{06}$ and $\hat{r}_{16}$ from statistical downscaling and the *Constant Reflectance Ratio Approach* is presented in Figures 4(a)–(b), respectively. For both $\hat{r}_{06}$ and $\hat{r}_{16}$ the majority of data points is positioned along the 1:1 line, and the correlation

coefficient is $R \approx 1.00$. The derived reflectances from the two independent approaches are very similar, and the probability density of the few larger deviations is several orders of magnitude below the maximum probability. There are a limited number of occurrences where $\hat{r}_{06}$ and $\hat{r}_{16}$ from the statistical downscaling approach are slightly larger than the ones from the *Constant Reflectance Ratio Approach*. However, since these samples are three to seven orders of magnitude less likely than the observations around the 1:1 line, they do not change the high correlation and slope of $1.00$. One minor difference between the

two results concerns the number of negative $\hat{r}_{16}$, which can occur for very thin clouds (i.e., very low $\tilde{r}_{\mathrm{HV}}$ and $\tilde{r}_{16}$). For the analyzed data set, almost all such observations are the result of the statistical downscaling technique with a relative contribution of $96.98\%$. However, the overall fraction of data points with a negative $\hat{r}_{16}$ is very low with a value of about $0.005\%$.

### 4.3 Lookup Table Approach

A third method to derive high–resolution cloud property retrievals for SEVIRI utilizes an iterative approach to determine $\delta r_{06}$

and $\delta r_{16}$ independently, based on the shape of the LUT, while constraining the observed $r_{\mathrm{eff}}$ to that of the baseline approach (i.e., simple trigonometric interpolation, which yields reflectances $\tilde{r}_{06}$ and $\tilde{r}_{16}$ that only resolve the large–scale variability. While the previous approaches can be implemented as a pre–processor outside the actual retrieval, this method requires access to the LUT and has thus been implemented through modifications of the CPP retrieval algorithm.

Again, a simple linear relationship between $\delta r_{\mathrm{HV}}$, $\delta r_{06}$ and $\delta r_{08}$ based on Eq.(2) is assumed:

$$\delta r_{\mathrm{HV}} = a \cdot \delta r_{06} + b \cdot \delta r_{08}, \tag{12}$$





where the fit coefficients $a$ and $b$ are determined from the same techniques as described in section 4.1. The variation $\delta r_{\mathrm{HV}}$ of the HRV channel is obtained from the observations following Eq.(2), while $\delta r_{08}$ is calculated as the difference between $r_{08}$ from high– and low–resolution optical thickness $\tau$ based on the functional relation $\mathcal{F}$ of the reflectances and cloud properties stored in the LUT (which motivates the name of this method). Therefore, $\delta r_{06}$ can be derived from:

$$
\begin{aligned}
5 \quad \delta r_{06} &= \frac{1}{a} \cdot (\delta r_{\mathrm{HV}} - b \cdot \delta r_{08}), \\
\hat{r}_{06} &= \tilde{r}_{06} + \delta r_{06}, \\
\delta r_{08} &= \mathcal{F}_{08}(\hat{\tau}, \hat{r}_{\mathrm{eff}}) - \mathcal{F}_{08}(\tilde{\tau}, \tilde{r}_{\mathrm{eff}}).
\end{aligned}
\tag{13}
$$

Note that the addition of $\delta r_{08}$ in the calculation of $\delta r_{06}$ helps to account for the noticeable increase in surface albedo of vegetation—like surfaces at $\lambda > 700\,\mathrm{nm}$ (i.e., the vegetational step). This should improve the estimation of $\delta r_{06}$ for thin clouds 10 (i.e., $\tau < 10$) and cloud–edge pixels. For the SWIR reflectance, instead of relying on the imperfect linear model in Eq.(6) or assumptions about the inhomogeneity index $H_{\sigma,16}$, the adjustment $\delta r_{16}$ is determined iteratively to conserve the coarse–resolution, pixel–level (i.e., $3 \times 3\,\mathrm{km}^2$) value of the effective droplet radius. If $\tilde{\tau}$ and $\tilde{r}_{\mathrm{eff}}$ are the cloud properties based on trigonometric interpolation, and $\hat{\tau}$ and $\hat{r}_{\mathrm{eff}}$ are the higher–resolution retrievals, which are derived from an inversion of the functional relationship ($\mathcal{F}$) between the high–resolution reflectances $\hat{r}_{06}$ and $\hat{r}_{16}$ following:

$$
15 \quad (\hat{\tau}, \hat{r}_{\mathrm{eff}}) = \mathcal{F}^{-1}(\tilde{r}_{06} + \delta r_{06}, \tilde{r}_{16} + \delta r_{16}),
\tag{14}
$$

then $\delta r_{16}$ can be determined as:

$$
\delta r_{16} = \mathcal{F}_{16}(\hat{\tau}, \hat{r}_{\mathrm{eff}} = \tilde{r}_{\mathrm{eff}}) - \mathcal{F}_{16}(\tilde{\tau}, \tilde{r}_{\mathrm{eff}}).
\tag{15}
$$

This implies that a positive or negative $\delta r_{06}$ is connected to a positive or negative $\delta r_{16}$ using the LUT to adjust the SWIR subpixel reflectance variations in such a way to be representative of the respective standard–resolution $\tilde{r}_{\mathrm{eff}}$. As a result, we do 20 not expect any improvement for the $r_{\mathrm{eff}}$ retrieval during the transition to smaller scales. Instead, we try to find a physically reasonable constraint for $\delta r_{16}$ to achieve a reliable retrieval of the higher–resolution $\hat{\tau}$, while retaining the accuracy of the standard–resolution retrieval of $\tilde{r}_{\mathrm{eff}}$.

The *LUT Approach* is illustrated in Figure 5(a), where an example SEVIRI liquid–phase LUT for a specific solar zenith angle ($\theta_0 = 40°$), sensor zenith angle ($\theta = 20°$), and relative azimuth angle ($\varphi = 60°$) is shown. Vertical dashed lines and values 25 below the grid denote fixed $\tau$, while the horizontal dashed lines and values right of the grid denote fixed $r_{\mathrm{eff}}$ in units of microns. The green dot highlighted by the capital letter "A" represents an example SEVIRI reflectance pair of approximately $\tilde{r}_{06} = 0.33$ and $\tilde{r}_{16} = 0.34$, which maps to $\tilde{\tau} = 8$ and $\tilde{r}_{\mathrm{eff}} = 12\,\mu\mathrm{m}$ (i.e., the retrieval result for the high–resolution reflectances from trigonometric interpolation). The red line highlights the $\tilde{r}_{\mathrm{eff}} = 12\,\mu\mathrm{m}$ isoline. The two horizontal, blue arrows indicate a positive ($\delta r_{06,1}$) and negative ($\delta r_{06,2}$) adjustment to $\tilde{r}_{06}$ based on Eq.(13). Without an adjustment to $\tilde{r}_{16}$, these newly derived higher– 30 resolution $\hat{r}_{06}$ map to significantly larger and lower effective droplet radii of about $\hat{r}_{\mathrm{eff}} = 29\,\mu\mathrm{m}$ and $\hat{r}_{\mathrm{eff}} = 5\,\mu\mathrm{m}$, respectively. The adjustments $\delta r_{16,1}$ and $\delta r_{16,2}$ simply assure that the prior effective radius retrieval is preserved (i.e., $\hat{r}_{\mathrm{eff}} = \tilde{r}_{\mathrm{eff}}$). Due to curvature of the lines of fixed $r_{\mathrm{eff}}$ given by the LUT, small deviations of the coarse–resolution average from $\tilde{r}_{16}$ can still occur.





Note that the *LUT Approach* requires a prior cloud phase retrieval (either from the lower–resolution or interpolated reflectances) to determine the correct LUT for either liquid water or ice.

## 4.4 Adjusted Lookup Table Approach

In order to improve the estimation of $\delta r_{16}$ in the *LUT Approach*, two modifications to the previous assumption are introduced in this section. The first one aims to provide a more realistic estimate of $\tilde{r}_{\mathrm{eff}}$ compared to the $3 \times 3\,\mathrm{km}^2$ result, which subsequently is used to determine $\delta r_{16}$. The value of $\tilde{r}_{\mathrm{eff}}$ is derived from adiabatic theory, which provides a physically sound relationship between the derived high–resolution cloud variables:

$$\hat{r}_{\mathrm{eff}} = \tilde{r}_{\mathrm{eff}} \left( \frac{\hat{\tau}}{\tilde{\tau}} \right)^a .\tag{16}$$

Based on observations, the study by Szczodrak et al. (2001) confirmed the value of $a = 0.2$ predicted by theory for marine stratocumulus, so this is the value also adopted here. This approach is illustrated in Figure 5(b), where the $\tilde{r}_{\mathrm{eff}}$ retrieval based on the interpolated reflectances at point "A" is indicated by the red $r_{\mathrm{eff}}$–isoline. During the first iteration step $\delta r_{06}$ is derived from Eq. (13) and $\delta r_{16} = 0$, which maps to $\hat{\tau}^1$ in the LUT (the exponent 1 indicates the first iteration step). This value is highlighted by the vertical, blue line. Based on Eq. (16) the corresponding, adiabatic $\hat{r}_{\mathrm{eff}}^1$ is calculated (highlighted by the horizontal, blue line). This value determines the adjustment $\delta r_{16}$. Note, that the resulting reflectances at point "B" do not exactly map to $\tilde{\tau}^1$ after the first iteration. As a result, multiple iterations are necessary to derive the final cloud properties. It has however been relatively simple to merge this iteration into the iterative retrieval loop of the CPP retrieval.

A second approach to improve upon the *LUT Approach* again utilizes the shape of the LUT to derive a local slope $S = \partial r_{16}/\partial r_{06}$ from the simulated LUT reflectances. The value of $S$ is calculated at the position denoted by $\tilde{\tau}$ and $\tilde{r}_{\mathrm{eff}}$. In the iterative CPP retrieval, this requires that both low– and high–resolution cloud properties are estimated during each iteration until convergence of both properties is achieved. This approach is illustrated in Figure 5(c). Again, the initial $\tilde{r}_{\mathrm{eff}}$ retrieval based on the interpolated reflectances at point "A1" is indicated by the red $r_{\mathrm{eff}}$–isoline. The slope $S_{\mathrm{A1}}$ at this position in the LUT is highlighted by the solid, blue line. Based on the derived slope and $\delta r_{06}$ from Eq. (13) the corresponding $\delta r_{16}$ can be calculated for each iteration step. Two additional examples for initial starting points ("A2" and "A3") and the respective slopes ($S_{\mathrm{A2}}$ and $S_{\mathrm{A3}}$) are also shown. These examples indicate the change in slope for different parts of the LUT. For small $\tilde{\tau}$, the slope $S_{\mathrm{A3}}$ become steeper, which leads to a larger adjustment $\delta r_{16}$. Meanwhile, for large $\tilde{\tau} > 30$ (for this specific viewing geometry and LUT) the $\tilde{\tau}$ and $\tilde{r}_{\mathrm{eff}}$–isolines are nearly orthogonal and the respective slope $S_{\mathrm{A2}}$ and $\delta r_{16}$ are close to $0$.

Both approaches introduced in this section have advantages and disadvantages, but promise to improve on the standard *LUT Approach*. While physically sound, adiabatic assumptions might not always be appropriate, especially for highly convective clouds or in the presence of drizzle. Meanwhile, large $\delta r_{06}$ adjustments might map to a point in the LUT where the derived local slopes at the position of $\tilde{\tau}^i$ and $\tilde{r}_{\mathrm{eff}}^i$ might not be representative anymore.



### 4.5 Comparison of interpolated and downscaled SEVIRI reflectances

In order to illustrate the difference between the various reflectances, a statistical comparison between the downscaled results for $\hat{r}_{06}$ and $\hat{r}_{16}$ and the observations at the native SEVIRI scale (i.e., $r_{06}$ and $r_{08}$) is shown in Figure 6. To allow for a pixel–to–pixel analysis, each $r_{06}$ and $r_{08}$ at the original horizontal resolution of $3 \times 3 \, \mathrm{km}^2$ is replicated to each of the 9 available subpixels

at the HRV channel resolution. To put the resulting differences into perspective, a comparison between the downscaled and interpolated high–resolution reflectances is also provided. Note that only the statistical downscaling and *Constant Reflectance Ratio Approach* are shown, because in the *LUT Approach* $\hat{r}_{06}$ and $\hat{r}_{16}$ are derived iteratively during the cloud property retrieval and are not provided as an output variable by the algorithm.

Figure 6(a) shows a PDF of the relative difference ($\Delta r_{06}$; shown in in red), which is defined as the difference between

$\hat{r}_{06}$ from the statistical downscaling approach and the resampled $r_{06}$, normalized by $r_{06}$, for an example SEVIRI scene from the Germany domain on 9 June 2013 at 10:55 UTC. Overall, $n = 696,879$ are included in the analysis. The distribution is centered around $\Delta r_{06} \approx 0$ and is almost symmetrical on both sides. The $1^{\mathrm{st}}$, $50^{\mathrm{th}}$, and $99^{\mathrm{th}}$ percentiles of $\Delta r_{06}$ are $-24.17\%$, $0.03\%$, and $27.85\%$, respectively. This means, that statistically the two different resolution yield similar reflectance observation, but high–frequency variability, which is resolved by $\hat{r}_{06}$, introduces significant deviations from the results at the standard

resolution. Overall, most of the observations, defined by the $25^{\mathrm{st}}$, $75^{\mathrm{th}}$ percentiles (i.e., $50\%$ of the data points), are in the range of $-3.12\%$ to $2.87\%$. These differences compare well to those observed for the downscaled $\hat{r}_{06}$ from the *Constant Reflectance Ratio Approach* (shown in blue). As expected, the relative differences between $\hat{r}_{06}$ and $\tilde{r}_{06}$ (shown in black) are visibly smaller. The $1^{\mathrm{st}}$ and $99^{\mathrm{th}}$ percentiles of $\Delta r_{06}$ are $-11.28\%$ and $12.54\%$, respectively, and most observations are in the range of $-1.43\%$ to $0.99\%$. As before, the distribution is centered around $\Delta r_{06} \approx 0$, with a median of $0.03$. The normalized root-mean-square

deviation (nRMSD; defined as the RMSD between $\hat{r}_{06}$ and $\tilde{r}_{06}$, normalized by the mean $\tilde{r}_{06}$) is $\mathrm{nRMSD} = 2.73\%$, which is less than half the value from the difference between $\hat{r}_{06}$ and the resampled $r_{06}$ ($\mathrm{nRMSD} = 6.30\%$).

A similar analysis for the channel 3 reflectances $r_{16}$, $\hat{r}_{16}$ (from Eq.(7)), and $\tilde{r}_{16}$ is shown in Figure 6(b). As before for the VNIR channel, the PDF of the relative differences ($\Delta r_{16}$) is centered around $\approx 0$, and the $1^{\mathrm{st}}$ and $99^{\mathrm{th}}$ percentiles are $-27.53\%$ and $28.85\%$ for the difference between $\hat{r}_{16}$ and the resampled $r_{16}$ and $-13.59\%$ and $11.40\%$ for the difference between $\hat{r}_{16}$

and $\tilde{r}_{16}$, respectively. The nRMSD is $3.16\%$ ($r_{16}$) and $6.24\%$ ($\tilde{r}_{16}$). Overall, $50\%$ of the data points lie in the range of $-2.67\%$ to $2.71\%$ (for the difference between $\hat{r}_{16}$ and $r_{16}$). Again, the results from the two downscaling approaches are very similar.

It has to be noted, however, that deviations of $\pm 3\%$ in the reflectances at the different spatial scales can have a significant impact on the remote sensing products of optical and microphysical cloud parameters, especially if the clouds are thin or the pixels are partially cloudy (Werner et al., 2018a, b). These impacts become even more pronounced for the samples with larger

deviations between downscaled and native reflectances. Such effects are illustrated in section 5.

### 5 Example retrievals

An example of a standard SEVIRI red, green, and blue (RGB) composite and the respective cloud property retrievals, utilizing the native $r_{06}$ and $r_{16}$, are shown in Figures 7(a)–(c). In comparison, the retrieval results using the downscaled $\hat{r}_{06}$ and $\hat{r}_{16}$





from the *Adjusted Lookup Table Approach*, using the *LUT Slope Adjustment*, are presented in Figures 7(d)–(f) for the same cloud field. The example is a $100 \times 100\,\mathrm{km}^2$–subscene of SEVIRI observations of an altocumulus field, which was acquired on 9 June 2013 at 10:55 UTC over ocean within the Germany domain. The three illustrated parameters are an RGB composite image of SEVIRI channel 3, 2, and 1 reflectance in panels a) and c), the cloud optical thickness $\tau$ and $\hat{\tau}$ in panels b) and e),

as well as the effective droplet radius $r_{\mathrm{eff}}$ and $\hat{r}_{\mathrm{eff}}$ in panels c) and f). For the cloud variables only liquid–phase pixels are shown. An increase in contrast and resolved cloud structures is visible in the higher–resolution RGB composite. Regarding the retrieved cloud properties, the fields of lower–resolution $\tau$ and $r_{\mathrm{eff}}$ are a lot smoother and the results exhibit less dynamical range than their higher–resolution counterparts. One obvious example is the bright cloudy part along $54.6°N$, where $\tau > 45$ are observed. Moreover, the region of low $r_{\mathrm{eff}}$ in the north–eastern corner of the scene exhibits more nuanced values in the

higher–resolution data set. Note, that for this case, the number of failed retrievals is reduced for the *Adjusted Lookup Table Approach* (see south–eastern corner of the scene).

## 6    Evaluation of downscaling techniques with MODIS data

This section presents an evaluation of the different downscaling techniques which are introduced in section 4, by means of MODIS observations. MODIS provides reflectances at a horizontal resolution of $1 \times 1\,\mathrm{km}^2$. These observations are re–mapped

to the higher–resolution grid of the SEVIRI $r_{\mathrm{HV}}$–band samples, and provide the means to derive reference retrievals of $\tau$ and $r_{\mathrm{eff}}$. Note, that even though these reference retrievals are performed at a higher resolution the "ˆ"–notation is omitted, because these cloud products are derived from actual observations, and are not the estimates obtained from the various downscaling techniques. Subsequently, the re–mapped, higher–resolution reflectances are smoothed using the spatial response function of the corresponding SEVIRI channels. The reader is reminded, that these data are still available at a higher resolution than the

native $3 \times 3\,\mathrm{km}^2$ grid of the SEVIRI $r_{06}$, $r_{08}$, and $r_{16}$ channels, but no longer contain any information about the high–frequency reflectance variability. As the simplest approach to derive higher–resolution cloud products, these results are called the baseline results. Subsampling also enables a comparison with SEVIRI's native 3 km observations.

These observations subsequently provide the means to apply the various downscaling techniques, as well as the simple triangular interpolation approach, in order to compare the retrieved cloud products (i.e., $\hat{\tau}$ and $\hat{r}_{\mathrm{eff}}$, as well as $\tilde{\tau}$ and $\tilde{r}_{\mathrm{eff}}$) to the

reference results. In addition, a comparison can be made to those cloud variables, which would be obtained at SEVIRI's native spatial resolution by setting each $3 \times 3$ pixel block to its central value.

Figure 8 shows RGB composites of the four example scenes, which comprise the data set for the evaluation of the different downscaling techniques. The scenes are increasingly more heterogeneous, starting with a rather homogeneous altocumulus field in Figure 8, two more heterogeneous broken altocumulus examples in Figures 8(b)–(c), and finally a broken cumulus field

in Figure 8(d).

Meanwhile, table 1 summarizes the ten different retrieval experiments that form the comparison in this section. For the sake of completeness, the reference data (i.e., the results from the re–mapped $1 \times 1\,\mathrm{km}^2$–reflectances) are also included. The cloud products derived from triangular interpolation of SEVIRI samples are referred to as the baseline data set, as this is the easiest





approach and any reliable downscaling technique needs to add an improvement on those results. These results are, however, not directly comparable with retrievals at SEVIRI's native 3 km resolution, which are added as a separate experiment and are obtained by sub–sampling the baseline results. Here, each central pixel of a $3 \times 3$ block is replicated nine times and compared to the 1 km reference. Experiments 1a and 1b denote the statistical downscaling approach from section 4.1. Here, 1a is based

on $\hat{r}_{06}$ and $\tilde{r}_{16}$ (i.e., only the VNIR reflectance is downscaled; the SWIR reflectance is derived from interpolation), while 1b utilizes both $\hat{r}_{06}$ and $\hat{r}_{16}$ (i.e., both reflectances are downscaled and thus include small scale reflectance variability). Similarly, retrievals based on the *Constant Reflectance Ratio Approach* and the *LUT Approach* are indicated as experiments 2a and 2b, as well as 3a and 3b, respectively. The retrievals from the two *Adjusted LUT Approaches* are denoted as experiments 3c and 3d.

First, the collocation and re–mapping procedure for the native MODIS reflectances is briefly described. A comparison

between the retrieved cloud products from the interpolation, as well as the different downscaling procedures, and the reference results follows in section 6.2. These retrievals can be used to derive estimates of the liquid water content ($W_\mathrm{L}$, $\tilde{W}_\mathrm{L}$, and $\hat{W}_\mathrm{L}$) and the droplet number concentration ($N_\mathrm{D}$, $\tilde{N}_\mathrm{D}$, and $\hat{N}_\mathrm{D}$), which are evaluated in section 6.3. While the downscaling of SEVIRI VNIR reflectances is based on their linear relationship to the observed high–resolution $r_\mathrm{HV}$, the downscaling of SWIR reflectances is based on a number of assumptions, which might induce large uncertainties in the retrieved cloud products.

Therefore, a comparison between the full downscaling techniques and the VNIR–only results is presented in section 6.4.

## 6.1 Reprojection of MODIS swath radiances to the SEVIRI grid

To obtain a reliable higher–resolution reference data set, MODIS level 1b swath observations (MOD021km) have been projected to the grid of the SEVIRI HRV reflectance observations, which corresponds to the Geostationary Satellite projection with a pixel resolution of $1 \times 1\,\mathrm{km}^2$. Initially, the native HRV grid is oversampled by a factor of three in each dimension (i.e.

the target grid has a 333m resolution), and nearest–neighbor interpolation is used for the projection. This oversampled field is subsequently filtered with the spatial response function of the HRV channel as given by (EUMETSAT, 2006), to remove high-frequency variability not resolved by the sensor and, in particular, the artifacts introduced by the nearest–neighbour interpolation technique. Finally, this field is downsampled, such that only each central pixel of a $3 \times 3$ block is retained to represent the $1 \times 1\,\mathrm{km}^2$–value.

To perform the subsequent experiments, a second set of level 1b radiances are generated, where the spatial variability is reduced to match that of the $3\,\mathrm{km}$–channels of Meteosat SEVIRI. This step again involves the filtering of the respective reflectance field with the channel–specific spatial response function of the lower–resolution SEVIRI channels (EUMETSAT, 2006). In addition, a band–pass filter has been constructed from the difference between the modulation transfer functions of the HRV and the $0.6\,\mu\mathrm{m}$ and $0.8\,\mu\mathrm{m}$ channels (weighted by the coefficients of a linear model; see Deneke and Roebeling, 2010).

This filter is used to extract the high–frequency signal of the HRV channel.

It should be noted that retrievals based upon these radiances will be different than those based upon the original MODIS C6 radiances, or from an absolutely accurate representation of the (hypothetical) truly observed, high–resolution SEVIRI samples. For one, it uses the linear model of Cros et al. (2006) and Deneke and Roebeling (2010) as a proxy for the HRV channel, thereby excluding a potentially significant source of uncertainty. Moreover, MODIS acquires these reflectances under different



viewing geometries (note that the true viewing angles are used in the CPP retrieval, so within the limits of plane–parallel radiative transfer, this effect is accounted for), and the spectral characteristics of the MODIS and SEVIRI channels are not entirely comparable. However, the goal of this study is to provide a consistent reference data set and retrievals from a single retrieval algorithm core. Statistical comparisons between the operational MODIS C6.1 and SEVIRI results, as well as the new

high–resolution SEVIRI products, are presented in the companion paper Deneke et al. (2019). Moreover, some interesting use cases are demonstrated in that study, which can benefit from an increase in the spatial resolution of the derived SEVIRI cloud parameters.

## 6.2 Results for $\tau$ and $r_{\text{eff}}$

Figure 9(a) shows a comparison of $\tau$ at the native SEVIRI resolution, and the reference $\tau$ at the 1 km scale for the example

cloud field in scene 2, which is shown as an RGB composite image in Figure 8(b). A total of over $13,000$ cloudy pixels (liquid phase) are located in this scene. While for small reference $\tau < 20$ there is a reasonable agreement between the two data sets, there is increased scatter around the 1:1 line (indicated by the gray, dashed line) for larger values of cloud optical thickness. For reference $\tau > 40$, a substantial underestimation of the 3 km–$\tau$ is observed, which yields a sizable contribution to the nRD of $15.8\%$. Figures 9(b)–(c) show similar scatter plots of $\tau$ and $\hat{\tau}$ from both experiment 2b and 3d, respectively. It is

obvious that the results from these two downscaling techniques improve the agreement to the reference retrievals significantly. The correlation between the data sets is increased and the nRD is strongly reduced to values of $1.182\%$ (experiment 2b) and $1.589\%$ (experiment 3d).

A similar comparison between the reference $r_{\text{eff}}$ and $r_{\text{eff}}$ at native SEVIRI resolution, as well as $\hat{r}_{\text{eff}}$ from the same down-scaling experiments, is presented in Figures 9(d)–(f). Here, the native–resolution results show a much better agreement with

the reference retrievals and, compared to the cloud optical thickness, the nRD$= 5.505\%$ is much lower. While experiment 2b exhibits a good agreement between reference $\tau$ and $\hat{\tau}$, the comparison of retrieved $\hat{r}_{\text{eff}}$ to the reference results is less favor-able. Both the reduced correlation ($R = 0.943$ versus $R = 0.964$), as well as the increased scatter around the 1:1 line (nRD $= 6.630\%$) indicate that the results from experiment 2b are less reliable than the ones performed at the native 3 km resolution. Thus, the elaborate downscaling procedure actually reduces the accuracy of the retrievals. In contrast, the retrieved $\hat{r}_{\text{eff}}$ from

experiment 3d improve upon the native–resolution results, with slightly better values of $R = 0.976$ and nRD $= 4.402\%$.

Statistics of the comparison between the reference and native 3 km, baseline, and experimental retrievals are presented in Figures 10(a)–(d) for example scenes 1–4, respectively. The parameters which are used to quantify the individual comparisons are the median of the relative difference (abbreviated with p50) to indicate the average deviation from the reference results, the interquartile range (IQR; defined as the relative difference between the $75^{\text{th}}$ and $25^{\text{th}}$ percentile of the deviation to the

reference retrievals) to indicate the spread between the different data sets, the nRD as a second measure of the spread of data points, and the explained variance ($R^2$, which equals the square of Pearson's product-moment correlation coefficient $R$) between the different retrievals and the reference. Values with a green and red background highlight the respective experiment with the best and worst comparison for the specific parameter. Yellow backgrounds, meanwhile, indicate all other experiments in between the two extreme results. The first noteworthy observation concerns the native and baseline retrievals of $\tau$, which



universally exhibit the largest median deviations and spread to the reference results as well as the lowest $R^2$. Still, the difference between native and baseline results indicates that the trigonometric interpolation to the HRV grid has significantly improved the comparison. For scene 2, the $1^{st}$, $50^{th}$, and $99^{th}$ percentiles of the absolute deviations of the native retrievals from the reference $\tau$ are $-13.54$, $-0.08$, and $6.96$, respectively. In contrast, each retrieval of $\hat{\tau}$ that accounts for small–scale reflectance variability,

yields significant improvements, regardless of the approach. This is especially obvious in the parameters that characterize the spread in the deviations, i.e., IQR and nRD, which are between 2–9 and 3–10 smaller for the various experiments and example scenes, respectively. Experiments 1b and 2b, as well as 3d, seem to achieve the best agreement to the reference retrievals. For the data set from experiment 3d the $1^{st}$, $50^{th}$, and $99^{th}$ percentiles of absolute deviations improve to $-0.30$, $0.13$, and $1.36$, respectively.

Regarding the effective droplet radius, the agreement between the native 3 km and baseline retrievals and the reference results is significantly better. It is worth pointing out that $\tilde{r}_{eff}$, obtained only by interpolating reflectances to the HRV grid, performs better than the native–resolution $r_{eff}$ retrieval for all scenes. As an example, the $1^{st}$, $50^{th}$, and $99^{th}$ percentiles of the absolute deviations between native and reference results for example scene 2 are $-1.29\,\mu m$, $0.18\,\mu m$, and $2.03\,\mu m$, respectively. The most reliable downscaling approach seems to be experiment 3d, which performs noticeably better than experiments 1b (note

the increased nRD and reduced $R^2$ for scene 3), 3c (overall worst performance for scenes 1 and 2), and 2b (increased spread and overall issues for the heterogeneous cloud field in scene 4). This indicates that the linear model in Eq.(6), presuming general adiabatic cloud conditions, or assumptions about a constant ratio of VNIR and SWIR reflectances are not adequate to estimate higher–resolution $\hat{r}_{16}$, at least not for certain cloud conditions. In the case of experiment 2b, this is understandable, since the technique was developed for partially cloudy pixels (Werner et al., 2018b). For experiment 3d, the $1^{st}$, $50^{th}$, and $99^{th}$

percentiles of the absolute deviations are comparable to the baseline data set, with values of $-0.30\,\mu m$, $0.13\,\mu m$, and $1.36\,\mu m$, respectively.

  The notably better performance of experiment 3d than 3b with respect to $\hat{r}_{eff}$ is somewhat surprising, and the specified goal that experiment 3b maintains the accuracy of the baseline $\tilde{r}_{eff}$ retrieval has not been fully reached. We believe that this might be caused by the sensitivity of the cloud property retrieval to small reflectance perturbations, in particular for broken clouds.

We plan to investigate this effect further in future studies.

### 6.3 Results for $W_L$ and $N_D$

Retrievals of $\tau$ and $r_{eff}$ (regardless of the resolution they are derived at) provide the means to infer other commonly used cloud variables. The $W_L$, which describes the amount of liquid water in a remotely sensed cloud column, can be derived as the product of retrieved cloud products (Brenguier et al., 2000; Miller et al., 2016):

$$W_L = \Gamma \cdot \rho_l \cdot \tau \cdot r_{eff}. \tag{17}$$

Here, $\rho_l$ and $\Gamma$ are the density of liquid water and a coefficient, which accounts for the vertical structure of the cloud profile ($\Gamma = 2/3$ for vertically homogeneous clouds, $\Gamma = 5/9$ for adiabatic clouds). Meanwhile, $N_D$ describes the number of liquid cloud droplets in a cubic centimeter of cloudy air. Calculating $N_D$ from remote sensing products requires a number of assump-





tions, which are summarized and discussed in Brenguier et al. (2000); Schüller et al. (2005); Bennartz (2007); Grosvenor et al. (2018). A simplified form of the resulting equation for $N_D$ is:

$$N_D = \alpha \cdot \tau^{0.5} \cdot r_{\text{eff}}^{-2.5}, \tag{18}$$

with $\alpha = 1.37 \cdot 10^{-5}$ (see Quaas et al., 2006). Note, that Eqs.(17)–(18) can yield both baseline and downscaled results (i.e.,
$\tilde{W}_L$ and $\tilde{N}_D$, as well as $\hat{W}_L$ and $\hat{N}_D$) when they are derived from the respective cloud optical thicknesses and effective droplet radii.

Similar to the comparison in section 6.2, scatterplots of the reference $W_L$, the native 3 km $\tilde{W}_L$ and the results from the downscaling experiments 1b and 3d ($\hat{W}_L$) are shown in Figures 11(a)–(c), respectively. As before, data is provided by example scene 2 sampled on 9 June 2013 at 10:55 UTC. Compared to the native SEVIRI results, a noticeable improvement in the
correlation and nRD is achieved by utilizing the two downscaling experiments. Not only are $\hat{W}_L$ closer to the 1:1 line, but the significant underestimation of the 3 km $W_L$ for larger reference results is mitigated. Especially for experiment 3d, the spread is less than one third the value of the baseline results (4.857% versus 15.234%). Regarding the comparison between reference and native $N_D$, as well as $\hat{N}D$, downscaling experiment 2b yields less favorable results. There is a slight decrease (increase) in $R$ (nRD). This is caused by the large IQR and nRD of the deviations in the retrieved $\hat{r}_{\text{eff}}$, shown in Figure 9(e), which are
amplified due to the associated power of 2.5 in Eq. (18). However, the derived values from experiment 3b are significantly better agreement with the reference $N_D$.

Values of p50, IQR, nRD, and $R^2$ for the $W_L$ and $N_D$ comparison from the four example scenes are illustrated in Figures 12(a)–(d). Due to the large deviations between the native $\tau$ and the reference retrievals, $W_L$ for the 3 km results almost universally show the largest deviations to the reference values, and thus the largest IQR and nRD, as well as the lowest explained
variance. The exception is the heterogeneous cloud field in the fourth example scene, where the large deviations between $\hat{r}_{\text{eff}}$ from experiment 2b and the reference retrievals yield the worst comparison for the respective $\hat{W}_L$. As for the statistical comparison in section 6.2, experiment 3c overall performs worst for scenes 1 and 2. However, 27 of the 32 comparisons exhibit the best results for experiment 3d. For the four example scenes considered in this analysis, it is obvious that the *Adjusted Lookup Table Approach*, using the *LUT Slope Adjustment*, is preferable to other downscaling approaches and yields more reliable
high–resolution cloud variables than the standard–resolution SEVIRI results.

For example scene 2, the the $1^{\text{st}}$, $50^{\text{th}}$, and $99^{\text{th}}$ percentiles of absolute deviations of the 3 km cloud variables from the reference $W_L$ are $-88.50\,\text{g}\,\text{m}^{-2}, 0.70\,\text{g}\,\text{m}^{-2}$, and $57.90\,\text{g}\,\text{m}^{-2}$, respectively, which for experiment 3d changes to $-15.55\,\text{g}\,\text{m}^{-2}$, $3.10\,\text{g}\,\text{m}^{-2}$, and $28.95\,\text{g}\,\text{m}^{-2}$. While a slight bias is introduced, the spread of deviations is significantly reduced. Meanwhile for $N_D$ these deviations are $-77.57\,\text{cm}^{-3}, -7.44\,\text{cm}^{-3}$, and $36.25\,\text{cm}^{-3}$ for the 3 km results and $-55.75\,\text{cm}^{-3}, -5.59\,\text{cm}^{-3}$,
and $27.04\,\text{cm}^{-3}$ for experiment 3d.

## 6.4   Full downscaling versus VNIR only

Apart from the *Constant Ratio Approach*, the downscaling of $r_{06}$ for each of the techniques presented in section 4 uses the well established relationship between $r_{06}$, $r_{08}$, and the averaged $\langle \tilde{r}_{\text{HV}} \rangle$ (see Figure 3 and the discussion in Deneke and Roebeling,





2010). In contrast, downscaling of $r_{16}$ is based on different assumptions about the microphysical structure and cloud hetero-geneity, which induces a level of uncertainty in the subsequent cloud property retrievals. To test whether assumptions about $r_{16}$ actually improve the retrieval of $\hat{\tau}$ and $\hat{r}_{\mathrm{eff}}$, this section presents retrievals that include the results from experiment 3d for $\hat{r}_{06}$ but do not include the respective downscaling schemes for $\hat{r}_{16}$. Instead, the SWIR reflectance for each sample is provided

by the $\tilde{r}_{16}$ value derived from trigonometric interpolation.

Figure 13(a) shows PDFs of the relative difference ($\Delta\tau$) between $\tilde{\tau}$ from the baseline test (black), as well as $\hat{\tau}$ retrieved from experiments 3a (blue) and 3d (red), and the reference results (i.e., distributions of the difference between the data sets, normalized by the reference $\tau$). Data is from example scene 2, shown in Figure 8(b), sampled on 9 June 2013 at 10:55 UTC. The largest differences to the reference retrievals are observed for the baseline results, which only account for the large–scale

reflectance variability of the cloud scene. Here, relative differences cover the range of $-20.44\% < \Delta\tau < 28.22\%$ (these values indicate the $1^{\mathrm{st}}$ and $99^{\mathrm{th}}$ percentile of $\Delta\tau$, respectively). The distributions for experiment 3d is noticeably thinner and these observed ranges are reduced significantly to $-2.33\% < \Delta\tau < 3.14\%$. The differences $\Delta\tau$ for experiment 3a look closer to the one from the full downscaling experiment. However, the maximum of the distribution around $\Delta\tau \approx 0$ is lower than from experiment 3d, and the $1^{\mathrm{st}}$ percentile is actually higher than from the baseline retrievals. Clearly, the downscaling of both

VNIR and SWIR reflectances is preferable for the retrieval of $\hat{\tau}$. For the effective droplet radius, the experiment comparison looks significantly different. Both relative differences $\Delta r_{\mathrm{eff}}$ based on the baseline and experiment 3d results exhibit a similar behavior and the full downscaling approach only yields small improvements on the retrievals from trigonometric interpolation. Conversely, $\Delta r_{\mathrm{eff}}$ from experiment 3a yields a noticeably larger spread and the retrievals become less reliable.

Regarding $\Delta W_{\mathrm{L}}$ and $\Delta N_{\mathrm{D}}$, the results using the complete downscaling approach yield the narrowest distributions, with

significantly smaller minimum and maximum deviations (up to a factor of $5.6$) compared to the VNIR–only downscaling technique. Compared to the baseline results the reliability of derived liquid water path from experiment 3d is also improved.

A summary of the performance of downscaling experiments 1a–3a (i.e., where only the VNIR reflectances are downscaled) compared to that of experiments 1b-3b (i.e., the full downscaling approaches) for all four example scenes is given in table 2. Here, the $1^{\mathrm{st}}$, $50^{\mathrm{th}}$, and $99^{\mathrm{th}}$ percentiles of the relative differences between $\hat{\tau}$ and $\hat{r}_{\mathrm{eff}}$ and the reference retrievals are listed.

An almost universal reduction in the biases is observed when both VNIR and SWIR reflectances are downscaled. These results provide strong evidence that simultaneous downscaling of the SWIR reflectances is essential for providing reliable higher–resolution retrievals of $\hat{\tau}$ and $\hat{r_{\mathrm{eff}}}$, as well as the subsequently calculated $\hat{W}_{\mathrm{L}}$ and $\hat{N}_{\mathrm{D}}$.

This result is likely also relevant for retrieving cloud properties at highest–possible resolution from other multi–resolution sensors such as MODIS, VIIRS and GOES–R: here, VNIR reflectances are generally available at highest spatial resolution,

while SWIR reflectances have a 2–4 times lower sampling resolution. Based on the previous results, smooth interpolation of the SWIR reflectances to the VNIR resolution cannot be recommended. Instead, downscaling approaches such as those presented in section 4 should be adopted to avoid a scale–mismatch in the spatial variability captured by the VNIR and SWIR channels, or equivalently, a degraded accuracy of the $r_{\mathrm{eff}}$–retrieval.





## 7 Conclusions

In this work, several candidate approaches to downscale SEVIRI channel 1–3 reflectances from their native horizontal resolution of $3 \times 3 \, \text{km}^2$ to the horizontal $1 \times 1 \, \text{km}^2$–scale of the narrowband HRV channel observations are evaluated. The goal is to identify a reliable downscaling approach to provide the means to resolve higher–resolution, subpixel reflectance and cloud
property variations, which are only resolved by reflectances from SEVIRI's coincident HRV channel.

Three different methods are presented and evaluated: (i) a statistical downscaling approach using globally determined fit coefficients based on bivariate statistics, (ii) a local approach that assumes a constant heterogeneity index for different scales (i.e., the *Constant Reflectance Ratio Approach*), and (iii) an iterative approach utilizing both global statistics and the shape of the SEVIRI LUT, while assuming a constant subpixel $\tilde{r}_\text{eff}$ (i.e., the *LUT Approach*). For the latter technique, two modifications
(by assuming adiabatic cloud conditions or by deriving local slopes within the LUT) are introduced, which avoid the constraint of a fixed $\tilde{r}_\text{eff}$.

The different downscaling approaches are evaluated using MODIS observations of four example cloud fields at a horizontal resolution of $1 \times 1 \, \text{km}^2$, which are obtained by re–mapping onto the higher–resolution SEVIRI grid, followed by an optional smoothing with the sensor spatial response function of SEVIRI. This approach has the benefit of providing a reference data
set to which the results for the different downscaling techniques can be objectively compared. Simply using trigonometric interpolation of radiances to the higher–resolution grid of the HRV channel (the baseline approach) provides a significant improvement in agreement with the reference dataset for $\hat{\tau}$ and $\hat{r}_\text{eff}$ compared to the native 3 km resolution results. It is shown that either downscaling approach yields reliable retrievals of $\hat{\tau}$ at the horizontal resolution of the SEVIRI HRV channel. These results compare noticeably better with the reference retrievals than the ones from the baseline approach. This improvement is
illustrated by a lower median absolute bias and spread (factor of 2–10), as well as a higher observed correlation between the data sets. Regarding $\hat{r}_\text{eff}$, the baseline results are found to be reliable. Depending on the cloud type, the various downscaling techniques exhibit a significantly worse agreement to the reference retrievals. For more homogeneous altocumulus fields, the *LUT Approach* with adiabatic assumptions seems inadequate, while for the more heterogeneous cloud fields the performance of the statistical downscaling technique and the *Constant Reflectance Ratio Approach* decreases noticeably. The reliability of
$\hat{r}_\text{eff}$ utilizing the *LUT Approach* with an adjustement based on the calculation of local slopes is comparable to the baseline results, and improves upon the results at the native 3 km resolution. Overall, a similar behavior is observed for the derived $\hat{W}_\text{L}$ and $\hat{N}_\text{D}$. Here, the *LUT Approach*, in combination with the use of local slopes, exhibits the best agreement to the reference results for 27 out of the 32 comparisons (i.e., four example scenes, two cloud variables, and four evaluation parameters). Based on these results, this method seems to be favorable compared to the other downscaling approaches. The results are preferable
to those obtained from the standard–resolution SEVIRI narrowband reflectances and pave the way for future higher–resolution cloud products by the MSG–SEVIRI imager. Especially for $\hat{\tau}$ and $\hat{W}_\text{L}$, these improvements are significant, as even the baseline results show deviations from the reference data set of up to $\approx 11$ and $\approx 70 \, \text{g m}^{-2}$ for the observed example scenes.

Each of the downscaling techniques utilizes a well established relationship between the observed reflectance from SEVIRI channels 1 and 2, as well as the one from the broadband HRV channel. To test the validity of the different assumptions for the



downscaling of the SWIR band reflectance, the reliability of VNIR–only downscaling approaches is compared to the corresponding full downscaling procedure. For the former, the $1 \times 1\,\mathrm{km}^2$–SWIR observations are provided by the baseline technique. An almost universally improved reliability of the retrieved cloud products is observed when both VNIR and SWIR reflectances are downscaled. This illustrates that, for reliable retrievals, all channels need to capture small–scale cloud heterogeneities at

the same scale. This implies that, for other multi–resolution sensors such as MODIS, VIIRS, and GOES–R ABI, downscaling approaches should also be adopted to avoid a scale–mismatch of resolved variability in the VNIR and SWIR channels.

    Naturally, these results require more evaluation with a larger data set to validate the reliability of the approach under different observational geometries and cloud situations. If a similarly good agreement to a set of reference retrievals is found for a broad range of different test scenes, a significant step towards higher–resolution SEVIRI cloud observations is achieved. If our

results are confirmed, such retrievals would be a significant improvement of SEVIRI's current standard–resolution retrievals. Meanwhile, more elaborate downscaling schemes could potentially improve upon the methods presented here. As an example, one possible improvement on the *Adjusted Lookup Table Approach* with adiabatic assumptions would be an explicit fit of the relationship in Eq.(16 from the native, lower–resolution variables. This might also reveal valuable insights into the validity of the adiabatic assumption commonly adopted in remote sensing (Merk et al., 2016). In addition, a comprehensive evaluation

of the benefits of the higher–resolution SEVIRI cloud products for the subsequent estimation of solar surface irradiance is planned. In particular, a comparison of satellite retrievals based on Greuell et al. (2013) with observations of a dense network of pyranometers following the approach of (Deneke et al., 2009) and (Madhavan et al., 2017) is planned, which will enable detailed studies of the effects of spatial and temporal resolution of satellite observations.

*Code and data availability.* The MODIS and MSG radiance data used as input to the CPP retrieval, the Python code used for their generation,

and the retrieval output are available from the authors on request, and will be made publically available through the *ZENODO* data repository for the final paper. The CPP software is copyrighted by EUMETSAT, and is not publically available.

*Author contributions.* Both authors have shaped the concept of this study and in particular refined the considered downscaling approaches through extensive discussions. HD implemented the processing of the high–resolution processing scheme including the different downscaling approachs. FW carried out the analysis of the output and wrote the initial draft manuscript, which was subsequently refined by both authors.

*Competing interests.* The authors declare no competing interests.

*Acknowledgements.* This study was carried out within the frame of the German collaborative project MetPVNet funded by the German Ministry of Commerce, grant number 0350009E. The use of MODIS data obtained from the Level–1 and Atmosphere Archive and Distribution System Distributed Active Archive Center, and the use of SEVIRI data distributed by EUMETSAT and obtained from the TROPOS satellite



archive is gratefully acknowledged. The lead author, Frank Werner, is now employed by the Jet Propulsion Laboratory, California Institute of Technology. This work was done as a private venture and not in the author's capacity as an employee of the Jet Propulsion Laboratory, California Institute of Technology. The authors thank Anja Hünerbein, Fabian Senf, Marion Schrödter–Homscheidt, and Michael Schwartz for comments on earlier drafts of this paper, which helped to improve the submitted version.



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

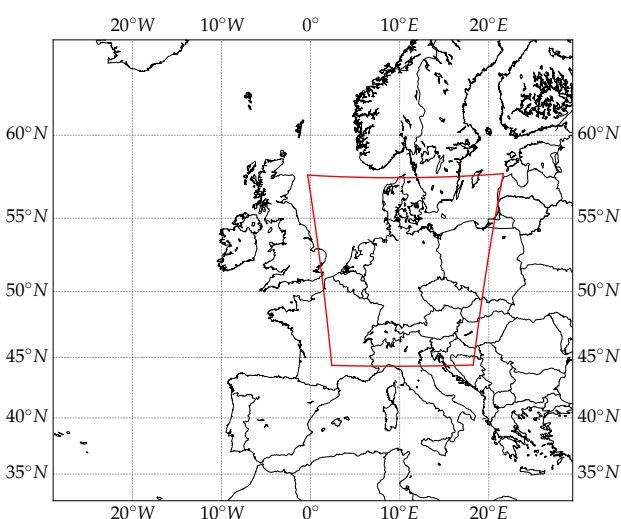

**Figure 1.** Map of the European SEVIRI domain, as defined in Deneke and Roebeling (2010). The red borders indicate the Germany domain, which is the focus of this study.





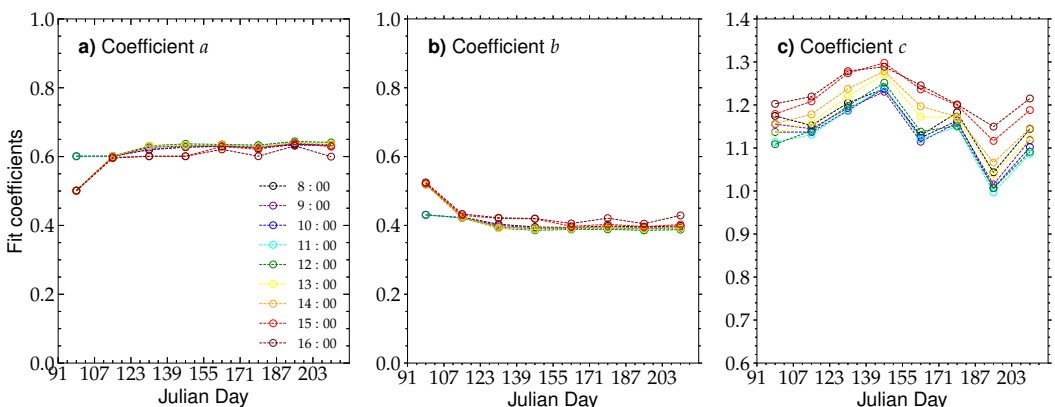

**Figure 2.** (a) Fit coefficients $a$, which are used to derive higher–resolution SEVIRI reflectances by means of statistical downscaling, as a function of Julian day. Coefficients are derived hourly and in 16–day intervals for the Germany domain between 1 April and 31 July 2013. Colors illustrate different UTC times. (b) Same as (a) but for fit coefficients $b$. (c) Same as (a) but for fit coefficients $c$.

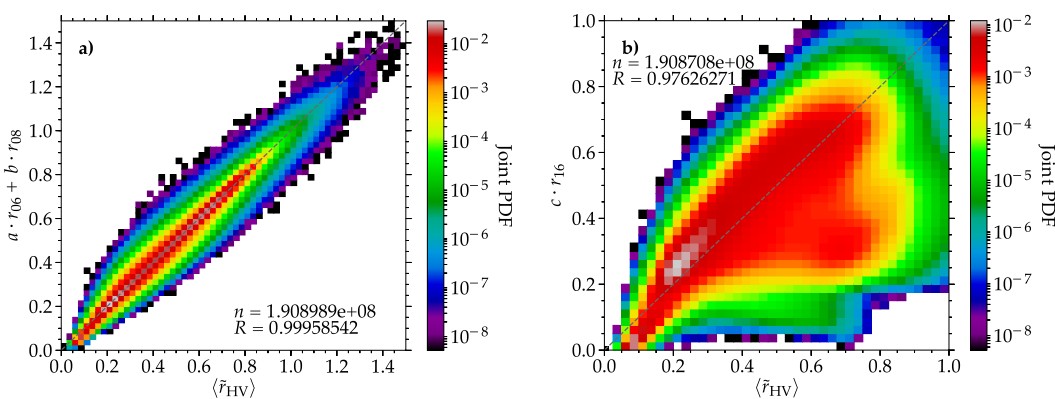

**Figure 3.** (a) Joint PDF of smoothed SEVIRI HRV reflectances ($\langle \tilde{r}_{HV} \rangle$) and those obtained from a linear model of observed SEVIRI channel 1 ($r_{06}$) and channel 2 reflectances ($r_{08}$), specifically $a \cdot r_{06} + b \cdot r_{08}$ (see section 4.1). Data is from all 5–minute SEVIRI observations of the Germany domain during June 2013. Only cloudy pixels are considered. The number of samples ($n$) and correlation coefficient ($R$) are given. (b) Same as (a) but for a linear model for SEVIRI SWIR reflectances, specifically $c \cdot r_{16}$.



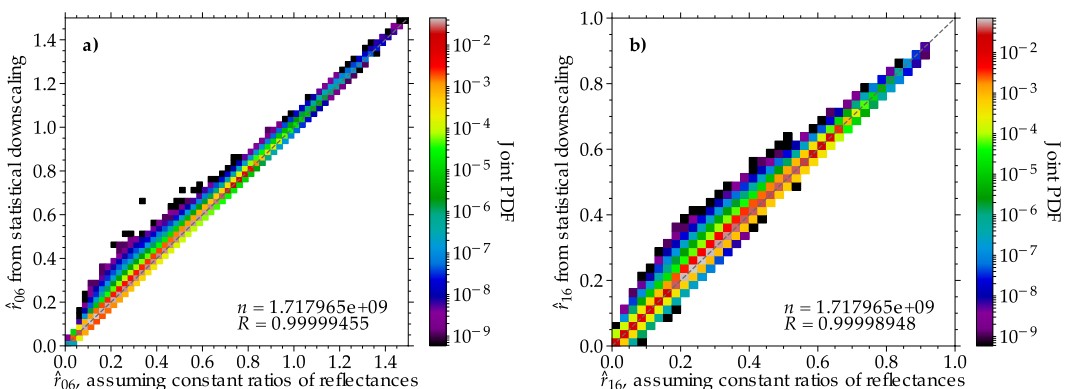

**Figure 4.** (a) Joint PDF of downscaled SEVIRI channel 1 reflectances ($\hat{r}_{06}$) from the *Reflectance Ratio Approach* (detailed in section 4.2) and those obtained from a linear model (described in section 4.1). Data is from all 5–minute SEVIRI observations of the Germany domain during June 2013. Only cloudy pixels are considered. The number of samples ($n$) and correlation coefficient ($R$) are given. (b) Same as (a) but for the comparison between downscaled SEVIRI channel 3 reflectances ($\hat{r}_{16}$) from the two downscaling techniques.



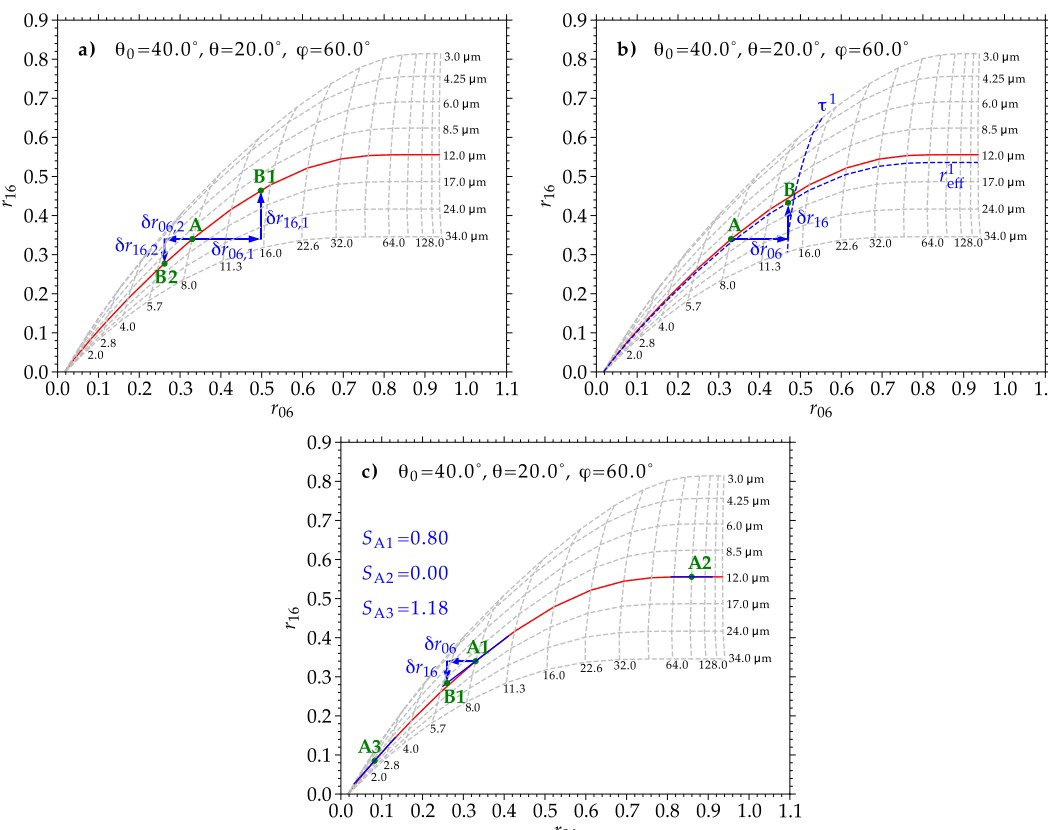

**Figure 5.** (a) Example SEVIRI lookup table for liquid–phase clouds, illustrating the *Lookup Table Approach* (introduced in section 4.3) for an observation highlighted by the reflectance pair indicated by point "A". For two different high–frequency variations of the channel 1 reflectance ($\delta r_{06,1}$ and $\delta r_{06,2}$) the derived high–frequency variations of the channel 3 reflectance ($\delta r_{16,1}$ and $\delta r_{16,2}$) is shown. See text for more description. (b) Same as (a) but illustrating the *Adjusted Lookup Table Approach* (introduced in section 4.4) with the *Adiabatic Adjustment* for a single $\delta r_{06}$ example. (c) Same as (b) but with the *LUT Slope Adjustment*.



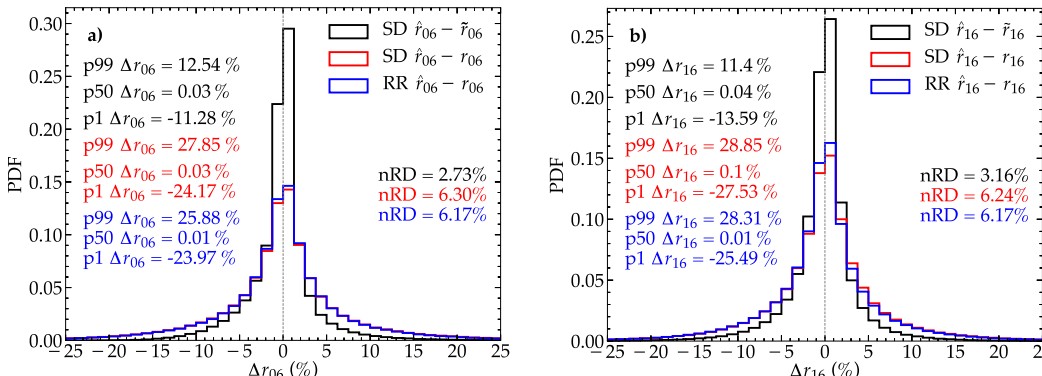

**Figure 6.** (a) PDF of the relative difference ($\Delta r_{06}$) between the downscaled SEVIRI channel 1 reflectances from the statistical downscaling approach (SD $\hat{r}_{06}$) and the higher–resolution channel 1 reflectances from trigonometric interpolation ($\tilde{r}_{06}$) is shown in black. Also shown is the relative difference between $\hat{r}_{06}$ from the statistical downscaling approach (SD; shown in red), as well as the *Constant Reflectance Ratio Approach* (RR; shown in blue), and the resampled original observations ($r_{06}$). Data is from SEVIRI observations of the Germany domain on 9 June 2013 at 10:55 UTC. Only cloudy pixels are considered. The $1^{\text{st}}$, $50^{\text{th}}$, and $99^{\text{th}}$ percentiles are given, as well as the normalized root-mean-square deviation (nRD; defined as the RD between the two data sets, normalized by the mean $\tilde{r}_{06}$ and $r_{06}$, respectively). (b) Same as (a) but for SEVIRI channel 3 reflectances ($r_{16}$, $\hat{r}_{16}$, and $\tilde{r}_{16}$).

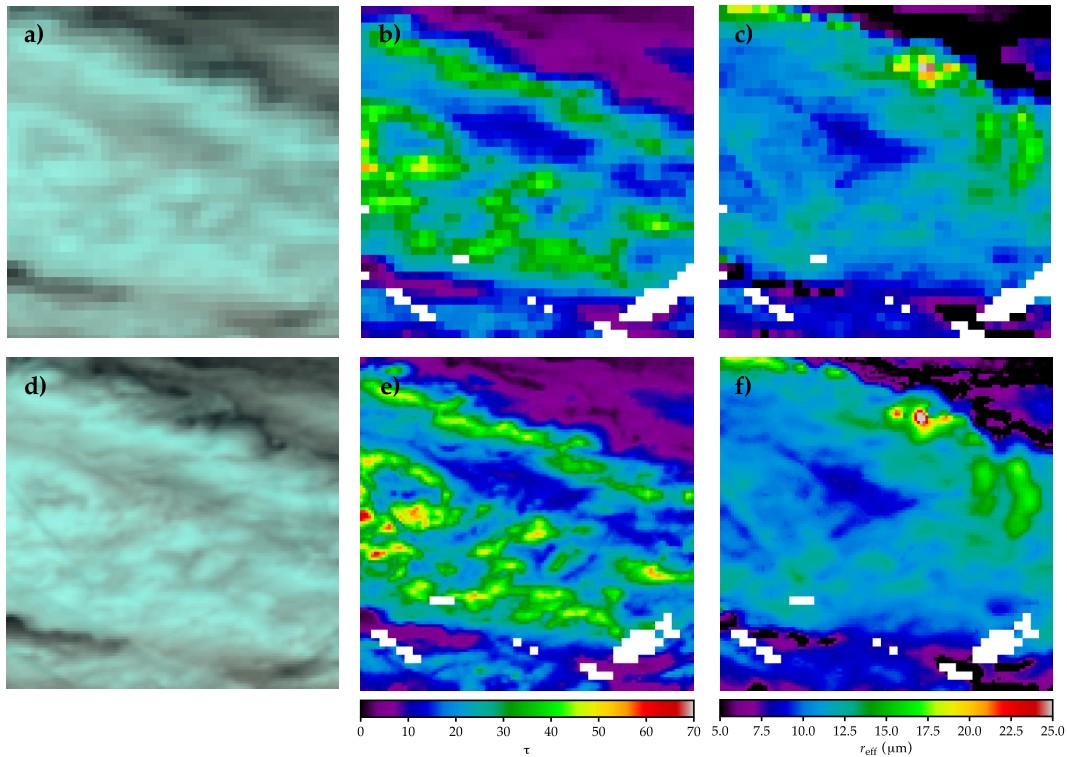

**Figure 7.** (a) RGB composite image of SEVIRI channel 3, 2, and 1 reflectances at the instrument's native horizontal resolution of $3 \times 3 \, \text{km}^2$. Data is from a $\approx 100 \times 100 \, \text{km}^2$ subregion within the Germany domain on 9 June 2013 at 10:55 UTC. (b) Similar to (a), but illustrating a map of the cloud optical thickness ($\tau$). White colors indicate pixel with either a failed cloud property retrieval, a non–liquid cloud phase, or non–cloud designation by the cloud masking algorithm. (c) Same as (b) but for the effective droplet radius ($r_{\text{eff}}$). (d)–(f) Same as (a)-(c) but at a horizontal resolution of $1 \times 1 \, \text{km}^2$. The reflectances and retrievals have been derived from the *Adjusted Lookup Table Approach* as described in section 4.4, using the *LUT Slope Adjustment*.

**a)** Scene 1: 20130601T1005Z

**b)** Scene 2: 20130609T1055Z

**c)** Scene 3: 20130605T1120Z

**d)** Scene 4: 20130606T1025Z

**Figure 8.** (a) RGB composite image of SEVIRI channel 3, 2, and 1 reflectances at the horizontal resolution of $1 \times 1 \, \mathrm{km}^2$ for example scene 1 sampled on 1 June 2013 sampled at 10:05 UTC. The reflectances have been derived from the *Adjusted Lookup Table Approach* as described in section 4.4, using the *LUT Slope Adjustment*. (b)–(d) Same as (a) but for example scenes 2 to 4, sampled on 9, 6, and 5 June 2013 at 10:55, 11:20, and 10:25 UTC, respectively.



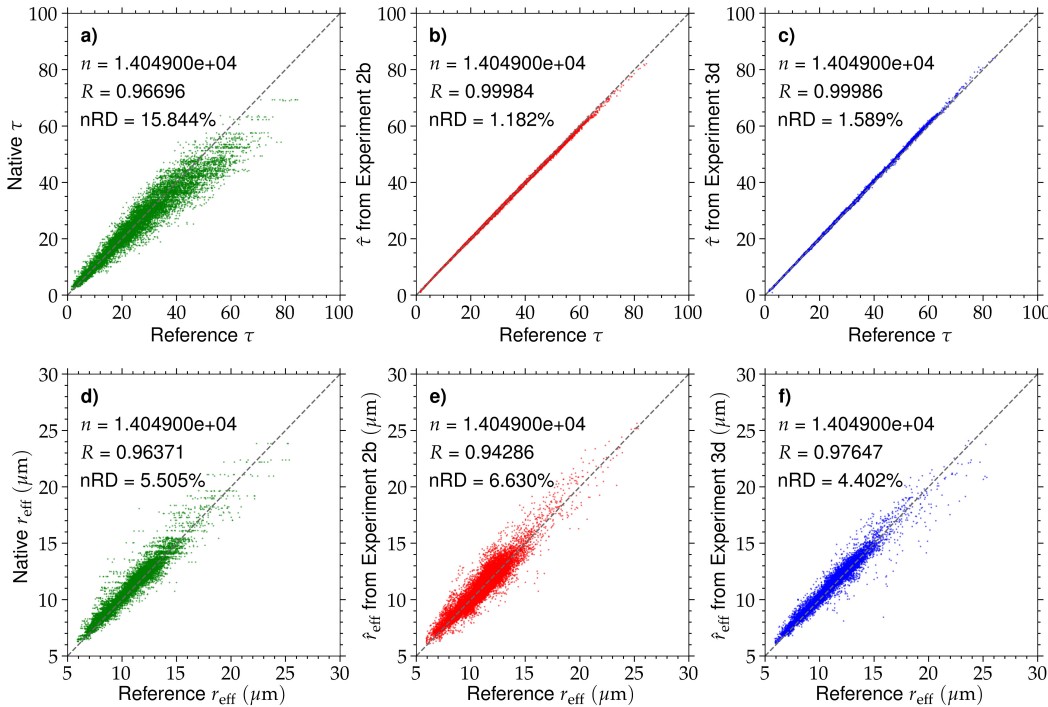

**Figure 9.** (a) Retrieved cloud optical thickness ($\tau$) from the native 3 km–retrieval as a function of the reference results ($\tau$ derived from the collocated MODIS reflectances at the $1 \times 1\,\mathrm{km}^2$ scale). Data is from example scene 2, sampled on 9 June 2013 at 10:55 UTC. The gray, dashed line represents the 1:1 line. The number of samples ($n$), correlation coefficient ($R$) and normalized root-mean-square deviation (nRD; defined as the RD between the two data sets, normalized by the average reference $\tau$) are given. (b)–(c) Same as (a) but for the comparison between $\tau$ and the results from experiments 1b and 3d ($\hat{\tau}$), respectively. (d)–(f) Same as (a)–(c) but for the effective droplet radius ($r_{\mathrm{eff}}$ and $\hat{r}_{\mathrm{eff}}$).





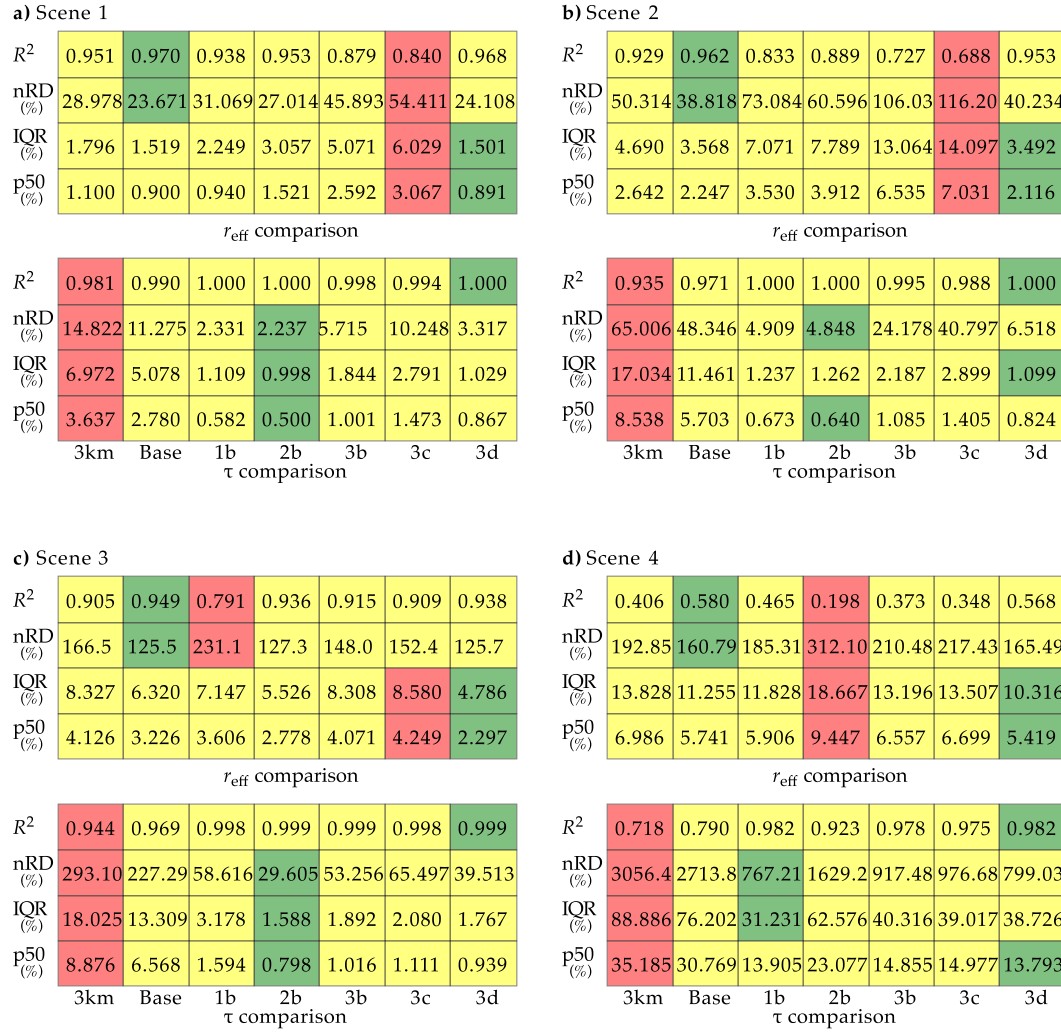

**Figure 10.** (a) Comparison of retrieved cloud optical thickness ($\tau$, bottom panels) and effective droplet radius ($r_{\mathrm{eff}}$, top panels) from the native 3 km resolution and baseline retrievals (i.e., only accounting for low–resolution reflectance variability), as well as the various downscaling experiments (1b, 2b, 3b, 3c, and 3d), and the reference retrieval results. Parameters to quantify the comparisons are the median of the relative difference to the reference (p50), relative interquartile range (IQR; $75^{\mathrm{th}}$-$25^{\mathrm{th}}$ percentile of the relative difference to the reference), normalized root-mean-square deviation (nRD; defined as the RD between the two data sets, normalized by the average reference retrieval), and the explained variance ($R^2$). Green colors indicate the experiment that compares best to the reference results, i.e., highest $R^2$ and lowest p50, IQR, and nRD. Red colors indicate the experiment with the worst agreement to the reference retrievals, while yellow colors indicate all experiments in between. Data is from example scene 1 sampled on 1 June 2013 sampled at 10:05 UTC. (b)–(d) Same as (a) but for example scene 2 to 4, respectively.



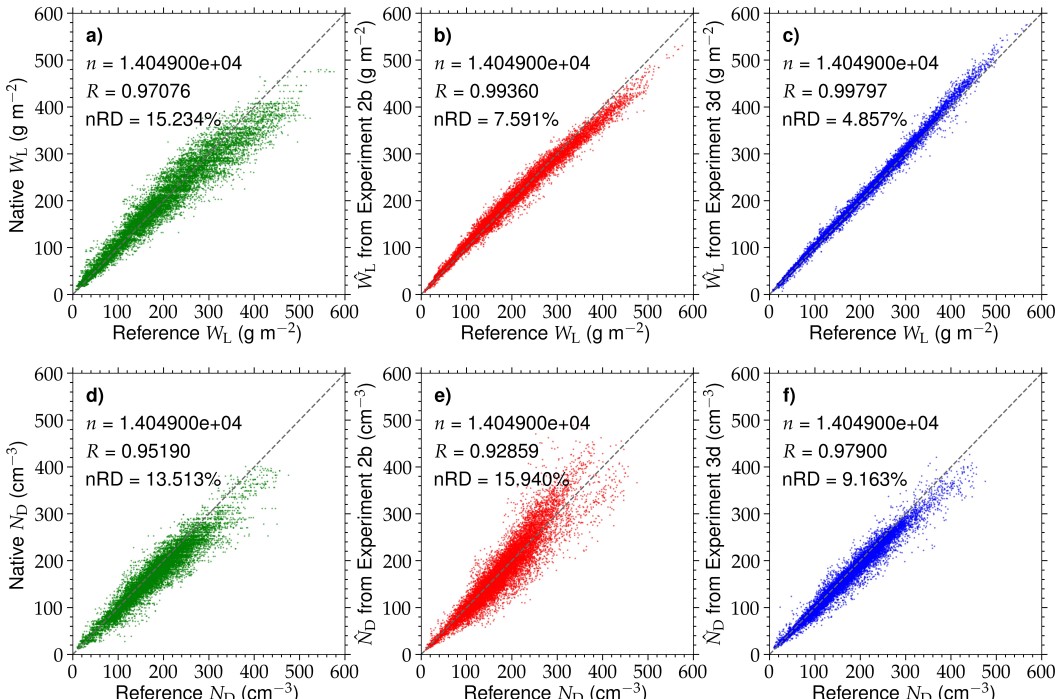

**Figure 11.** (a) Retrieved liquid water path ($W_L$) from the native 3 km resolution retrieval as a function of the reference results ($W_L$ derived from the colocated reflectances at the $1 \times 1\,\mathrm{km}^2$ scale). Data is from example scene 2, sampled on 9 June 2013 at 10:55 UTC. The gray, dashed line represents the 1:1 line. The number of samples ($n$), correlation coefficient ($R$) and normalized root-mean-square deviation (nRD; defined as the RD between the two data sets, normalized by the average reference $W_L$) are given. (b)–(c) Same as (a) but for the comparison between reference $W_L$ and the results from experiments 1b and 3d ($\hat{W}_L$), respectively. (d)–(f) Same as (a)–(c) but for the effective droplet radius. ($N_D$ and $\hat{N}_D$).



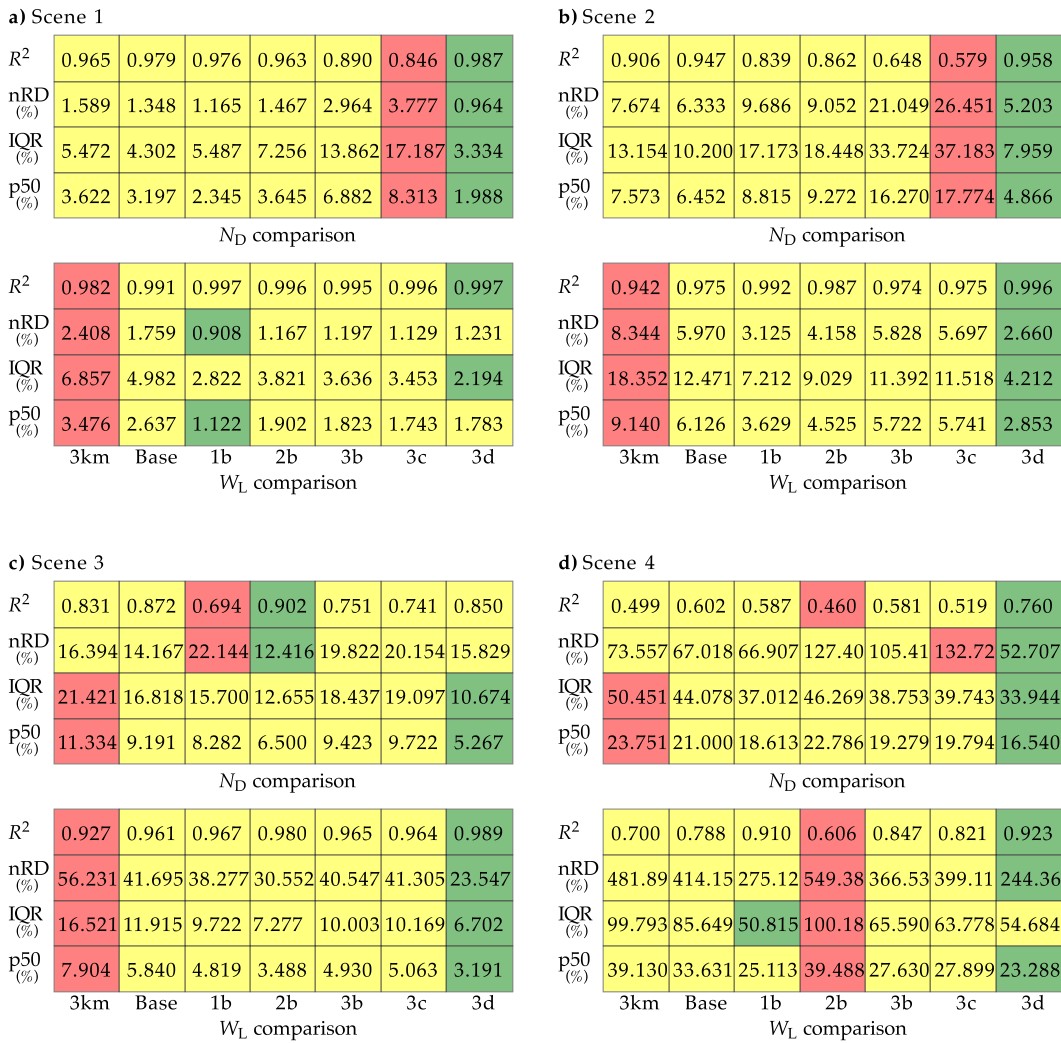

**Figure 12.** (a) Comparison of derived liquid water path ($W_L$, bottom panels) and droplet number concentration ($N_D$, top panels) from the native 3 km resolution and baseline retrievals, as well as the various downscaling experiments (1b, 2b, 3b, 3c, and 3d), and the respective reference results. Parameters to quantify the comparisons are the median of the relative difference to the reference (p50), relative interquartile range (IQR; $75^{th}$-$25^{th}$ percentile of the relative difference to the reference), normalized root-mean-square deviation (nRD; defined as the RD between the two data sets, normalized by the average reference retrieval), and the explained variance ($R^2$). Green colors indicate the experiment that compares best to the reference results, i.e., highest $R^2$ and lowest p50, IQR, and nRD. Red colors indicate the experiment with the worst agreement to the reference retrievals, while yellow colors indicate all experiments in between. Data is from example scene 1 sampled on 1 June 2013 sampled at 10:05 UTC. (b)–(d) Same as (a) but for example scenes 2 to 4, respectively.

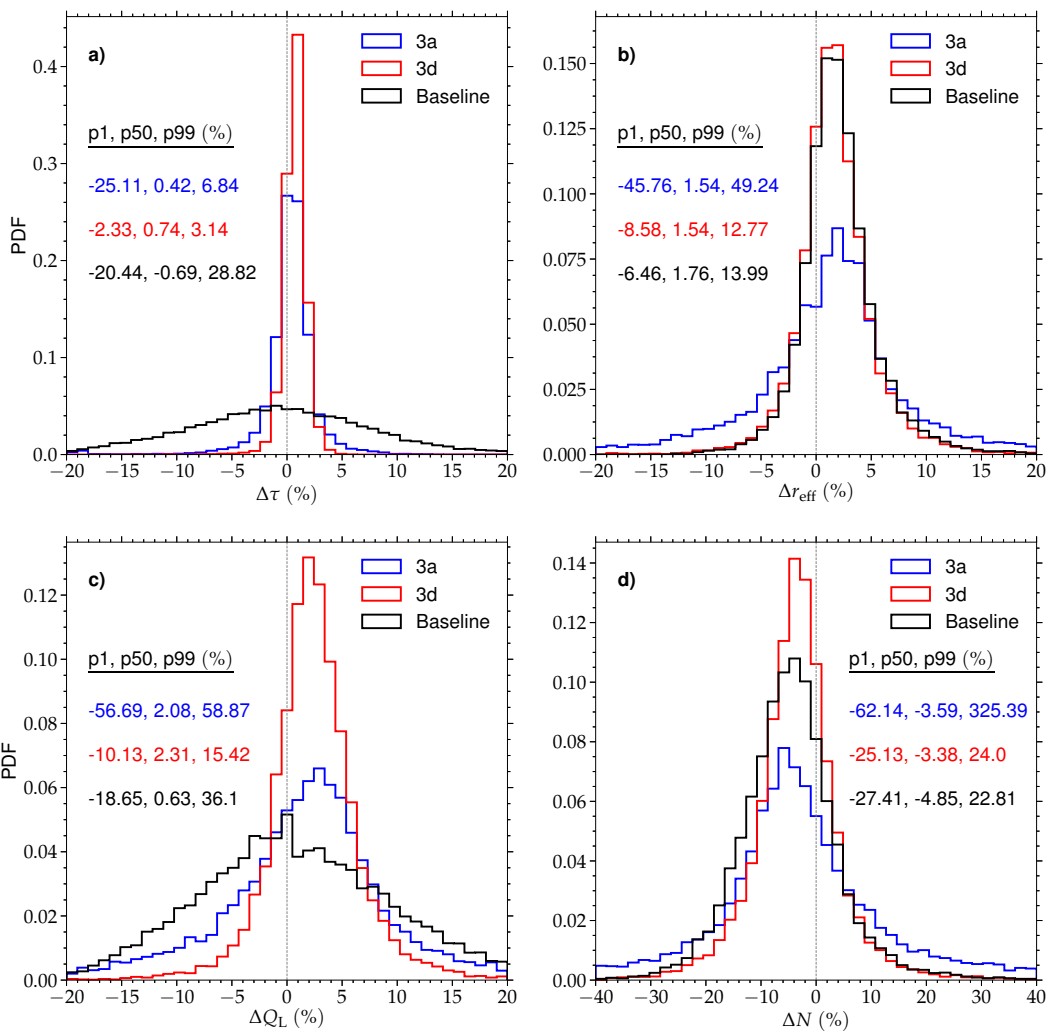

**Figure 13.** (a) PDFs of the relative differences ($\Delta\tau$) between the retrieved cloud optical thickness ($\tau$) from various downscaling methods (i.e., the baseline test, as well as experiments 3a and 3d, shown in black, blue, and red color, respectively) and the reference results (i.e., the original 1 km–retrievals). Data is from example scene 2 sampled on 9 June 2013 at 10:55 UTC, which is shown in Figure 8(b). The $1^{st}$, $50^{th}$, and $99^{th}$ percentiles of $\Delta\tau$ for each experiment are given. (b) Same as (a) but for $\Delta r_{eff}$, which is the relative difference for the retrieved effective droplet radius ($r_{eff}$). (c) Same as (a) but for $\Delta W_L$, which is the derived liquid water path ($W_L$). (d) Same as (a) but for $\Delta N_D$, which is the relative difference for the derived droplet number concentration ($N_D$).





**Table 1.** Description for the different retrieval experiments, which are characterized by different assumptions for the downscaling of SEVIRI reflectances from the native horizontal resolution of 3 km to the MODIS–like 1 km scale.

| Experiment | Description |
|---|---|
| Reference | $r_{06}$ and $r_{16}$ from the native $1 \times 1\,\mathrm{km}^2$ MODIS scale, collocated on the higher–resolution SEVIRI grid |
| Baseline | $\tilde{r}_{06}$ and $\tilde{r}_{16}$ from triangular interpolation, thus only accounting for low–frequency variabilities |
| Native 3 km | $\tilde{r}_{06}$ and $\tilde{r}_{16}$ subsampled to native SEVIRI grid, and each central value repeated $3 \times 3$ times |
| 1a | $\hat{r}_{06}$ from *Statistical Downscaling Approach* as described in section 4.1; $\tilde{r}_{16}$ from trigonometric interpolation |
| 1b | $\hat{r}_{06}$ and $\hat{r}_{16}$ from *Statistical Downscaling Approach* as described in section 4.1 |
| 2a | $\hat{r}_{06}$ from *Reflectance Ratio Approach* as described in section 4.2; $\tilde{r}_{16}$ from trigonometric interpolation |
| 2b | $\hat{r}_{06}$ and $\hat{r}_{16}$ from *Reflectance Ratio Approach* as described in section 4.2 |
| 3a | $\hat{r}_{06}$ from *Lookup Table Approach* as described in section 4.3; $\tilde{r}_{16}$ from trigonometric interpolation |
| 3b | $\hat{r}_{06}$ and $\hat{r}_{16}$ from *Lookup Table Approach* as described in section 4.3 |
| 3c | $\hat{r}_{06}$ and $\hat{r}_{16}$ from *Adjusted Lookup Table Approach* as described in section 4.4 (with adiabatic assumption) |
| 3d | $\hat{r}_{06}$ and $\hat{r}_{16}$ from *Adjusted Lookup Table Approach* as described in section 4.4; (with slope) |



**Table 2.** Comparison of the cloud property retrieval results from the downscaling experiments 1a–3a, which only account for the VNIR part, and experiments 1b–3b, which include adjustments to both VNIR and SWIR reflectances. The comparison shows the $1^{st}$, $50^{th}$, and $99^{th}$ percentiles of the relative differences $\Delta\tau$ (for the cloud optical thickness $\tau$) and $\Delta r_{eff}$ (for the effective droplet radius $r_{eff}$), which illustrate the deviation of the different retrieval approaches from the reference results, normalized by the reference retrievals. Data is from the four example scenes shown in Figure 8.

| Scene | $\tau$ (%) | | | | | | $r_{eff}$ (%) | | | | | |
|---|---|---|---|---|---|---|---|---|---|---|---|---|
| | 1a | 1b | 2a | 2b | 3a | 3b | 1a | 1b | 2a | 2b | 3a | 3d |
| **#1** | | | | | | | | | | | | |
| $1^{st}$ | -4.26 | -2.59 | -3.13 | -1.97 | -3.47 | -1.77 | -13.20 | -5.47 | -12.23 | -5.08 | -12.55 | -5.19 |
| $50^{th}$ | 0.28 | 0.19 | 0.16 | 0.00 | 0.52 | 0.81 | 0.82 | 0.11 | 0.81 | 0.00 | 0.85 | 0.76 |
| $99^{th}$ | 4.57 | 2.95 | 3.49 | 2.18 | 4.13 | 2.86 | 8.38 | 6.04 | 16.57 | 6.99 | 16.94 | 6.11 |
| **#2** | | | | | | | | | | | | |
| $1^{st}$ | -26.81 | -19.77 | -24.30 | -2.64 | -25.11 | -2.33 | -47.37 | -28.00 | -45.51 | -12.69 | -45.76 | -8.58 |
| $50^{th}$ | 0.45 | 0.30 | 0.21 | 0.11 | 0.42 | 0.74 | 1.51 | 0.57 | 1.49 | 0.50 | 1.54 | 1.54 |
| $99^{th}$ | 8.31 | 4.31 | 6.29 | 2.66 | 6.84 | 3.14 | 53.23 | 18.16 | 48.42 | 19.19 | 49.24 | 12.77 |
| **#3** | | | | | | | | | | | | |
| $1^{st}$ | -37.33 | -32.08 | -33.95 | -25.00 | -33.65 | -20.68 | -66.37 | -45.46 | -65.10 | -25.71 | -64.53 | -23.25 |
| $50^{th}$ | 0.00 | 0.06 | 0.00 | 0.00 | 0.25 | 0.38 | 0.74 | 0.43 | 0.47 | 0.00 | 0.53 | 0.49 |
| $99^{th}$ | 38.16 | 31.44 | 36.09 | 23.98 | 36.05 | 25.56 | 126.95 | 59.57 | 116.87 | 33.31 | 118.90 | 42.76 |
| **#4** | | | | | | | | | | | | |
| $1^{st}$ | -73.33 | -69.72 | -60.18 | -61.45 | -72.27 | -69.12 | -52.53 | -36.01 | -50.35 | -48.35 | -51.02 | -33.00 |
| $50^{th}$ | 2.86 | 1.47 | 1.23 | 7.07 | 2.71 | 2.61 | -0.13 | 0.00 | -0.13 | 0.26 | -0.13 | 0.13 |
| $99^{th}$ | 309.98 | 286.63 | 292.25 | 414.23 | 308.35 | 284.50 | 175.70 | 42.16 | 126.24 | 102.89 | 167.46 | 37.54 |