# Peer review of "Increasing the spatial resolution of cloud property retrievals from Meteosat SEVIRI by use of its high–resolution visible channel: Evaluation of candidate approaches with MODIS observations"

_Atmospheric Measurement Techniques, 2019_

## Referee Comment (RC1) · Anonymous Referee #1 · 6 Nov 2019

Review of "Increasing the spatial resolution of cloud property retrievals from Meteosat SEVIRI by use of its high–resolution visible channel: Evaluation of candidate approaches with MODIS observations" by Werner and Deneke.

The manuscript discusses a topic relevant for the scientific community. Various down-scaling techniques are presented and analysed in order to derive high resolution (1km) cloud properties from low resolution (3km) Meteosat SEVIRI data. Methods and mo-

tivations are layed out in detail, but the presentation suffers from the dizzying number of approaches used for downscaling, used for verification, of data spatial resolutions and of acronyms used. While I have full confidence in the authors scientific rigour, I was close to giving up reading through all the details offered. At the same time, I'm missing general interpretation of the large variety of results in some places. This leads to the unsatisfying point that the support of final and most important conclusions is not easy to find for the reader. I have the impression that this manuscript could be much improved by a major revision and tightening of the presentation, especially of the comparison results section 6.

Major points:

Especially section 6 is confusing. I would suggest to reduce the number of downscaling code versions. Especially the results section 6 even has versions not discussed anywhere. I would also suggest to reduce the number of error quantities discussed, maybe to the set shown in the tables Fig 10 and 12. Do not discuss other additional numbers in the text. Please see details below.

Is Deneke and Roebeling 2010 the basis of this paper? I had the impression down to page 8 that many things come from this older publication. If this is the case, it would be one possibility for shortening. You have to make the connection of the two clearer in the introduction.

Technical problem is that a companion paper (Deneke et al 2019) is obviosuly not submitted at this stage. I would recommend removal of all references to it or waiting for its puplication in discussion stage.

Specific points

p.1, l. 11 ff: Where do these numbers for tau, reff and WL and ND come from? I can not easily find these numbers in the manuscript and I hardly can find any discussion of them. Please extend discussion of these later on or remove them from the abstract.

[Figure]

p.1, l. 20: The whole abstract reads as if it does not work very well. Maybe apart from tau. This concluding sentence reverses these statements. Please revise.

p. 3, l. 11: Isn't a clear reference to Deneke and Roebeling 2010 missing here? Can you please make that clear in the introduction.

p. 3, l. 20: This way you will only get relative errors. All problems retrievals at 1km resolution still suffer from are not discussed or improved ... e.g. Zhang et al, 2009, 2011. Can you please mention that.

p. 3, introduction in general: Can you please make clearer: What is the motivation for an improved resolution of products? What problem do you expect to improve?

p. 4, l. 30: "horizontal resolution of 3 x 3". This is the nominal sub-satellite resolution. Can you please make clear whether you consider the full spatial response function for each point. Much later it sounds like, but here you widely stick to the simplified "1 km", "3 km" without further explanation.

p. 5, l. 14/15 and 17: I do not understand the need for theses statements. These are purely technical, aren't they? First you mention a method not used in this study?! With modifications described in a study not published yet?! Then you are talking about a version control system development branch (?) to make clear that this algorithm is not perfectly the same as in the unpublished companion paper!? I doubt that the reader needs these documentation details.

p. 5, l. 28: Again the reader wants to know whether you consider the different spatial response function of normal and high resolution channels? Please comment why you think you do not need to consider this or how it is considered.

p. 6, l. 16: "Statistical downscaling": I do not see the "statistical" element? Are not all downscaling steps fully deterministic? No random element is in there? Please clarify.

p. 7, l. 2: IQR=0.03 of what? Daily values? Hourly?

[Figure]

p. 7, l. 4, Fig 2: I can only see three colors for 14:00, 15:00 and 16:00 UTC?!

p. 8, l. 7: "Diurnally, the variability in the hourly derived ..." : You mean the IQR is derived over 18 or 19 hourly values over one day? Or over 16 days?

p. 10, eq. 9: Under which assumption does eq. line 2 follow from eq. line 1?

p. 10, l. 15, "R $\sim$ 1.0" : This proves that both approaches are rather equivalent, but suffer from the same core problem. the reff impact. Is this an important comparison or just a distracting sideshow?

p. 10, l. 22: I can see that the first approach could produce negatives, but the second?

p. 13, l. 1: Is this section including Fig 6 really needed? It confirms stuff that could be seen before and adds another side aspect.

p. 14, l. 11: You did never mention failed retrievals and reasons for it. Skip this sentence?

p. 14, l. 12, chapter: This section is confusing. I started reading with the understanding that you only use MODIS data in this chapter until I read the Fig. 8 caption which sounds like it shows SEVIRI data. Please make sure that this stays clear from the beginning and throughout the section. Do you use "SEVIRI data" or only "SEVIRI-resolution MODIS data" in this section?

p. 14, l. 18: You mention spatial response for the first time here I guess. What about spectral differences between MODIS and SEVIRI? Please discuss.

p. 14, l. 33, "interpolation of SEVIRI samples": Are you talking about SEVIRI or MODIS data here? See point above.

p. 15, l. 4-8: Phew! Now you add sub-experiments "a, b, c, d" on top of the new nomenclature "1, 2, 3 " ... I'm struggling, to keep reading ... At least, do not use "1,2,3" acronyms alone, but write out the experiments to make them more recognizable. Please do not introduce experiments you will not even discuss (1a, 2a).

p. 15, l. 11-12: And now ... a few new products on top. You have to mention the relevanve of these right in the introduciton.

p. 15, l. 21: This is not the first time you use the exact spatial response functions, isn't it? This is rather late to mention the reference for the first time.

p. 15, l. 23, "3x3 block". This block is 333 m resolution here, right? Please make sure that this can not be confused with the other 3x3 blocks mentioned before.

p. 15, l. 25, "level 1b": Could be easily confused with your experiment notation. You did not use the term level 1b data before, you do not need it here.

p. 15, l. 28-29: What is the "modulation transfer function" good for? Why do you only mention it here, that late in the manuscript?

p. 16, l. 2, "spectral characteristics": This is again too late to mention such an obvious problem that late.

p. 17, l16ff: I'm missing this kind of more conclusive interpretation elements instead of adding number to number in the text.

p. 17, l.31: This sounds as if 2/3 and 5/9 are magic numbers found empirically. Approx. 2/3 follow from optical properties directly and in general. 5/9 contains an empirical element. I would prefer to say "WL=2/3 ro tau reff" and adiabatic clouds have a typical additional factor of factor=5/6 due to their vertical structure.

p. 18, l. 3, eq. 18: Again, this equation seems to contain magic, but is rather simple in its core. Maybe give some additional explanation: "Droplet number could simply be derived from LWC and a droplet size. Using empirical factors accounting for typical droplet size distributions and vertical cloud structure, the following can be derived:"

p. 18, l. 22, "3c overall performs worst": Why? Can you give a general explanation or guess?

p. 19, l. 25, "results provide strong evidence that simulateneous downscaling of the

SWIR reflectances is essential": Again, why? Can you give a general explanation or guess?

p. 20, chapter: The conclusions section also needs more of this kind of general explanations and interpretations instead of repeating "x is better than y, so use x".

p. 21, l. 5/6: Many studies show that going below the 1 km scale might introduce new problems with variability which are smoothed out at this scale. Please discuss this caveat when making such a suggestion.

Fig. 9, caption: Here "1b" is mentioned. Isn't it "2b" in the text?

Fig 10.: Please write out the experiments in words in addition to number codes.

Fig. 6 and 13.: Needed?

---

## Referee Comment (RC2) · Anonymous Referee #2 · 24 Nov 2019

My opinion is that this manuscript presents significant work well worth publishing. The key achievement lies the development and testing of methods for using geostationary satellite data to obtain cloud properties at a three times higher spatial resolution than the current standard. The methodology is sound, and the presentation is generally clear. I recommend a number of minor refinements (mainly to improve clarity), but there is one issue I'd like to single out in particular.

[Figure]

The text says throughout the manuscript (starting with Lines 3-4 of the abstract) that the proposed methods can increase the spatial resolution of SEVIRI cloud products from 3 km to 1 km (from the resolution of most SEVIRI bands to the resolution of the SEVIRI HRV band). My understanding, however, is that the resolution of SEVIRI observations is 3 km and 1 km only at the sub-satellite point, and that this resolution degrades with the cosine of the viewing zenith angle. (See, for example, http://www.esa.int/esapub/bulletin/bullet111/chapter4_bul111.pdf or http://www.icare.univ-lille1.fr/projects/seviri-aerosols.) For the test area around Germany, this can increase the meridional extent of SEVIRI pixels by 40% or more. For the most part, considering this effect would require only a clarification in the text; the only part where this becomes a substantial issue is the comparison with MODIS data. Considering that the meridional resolution of MODIS images should remain around 1 km even if the SEVIRI resolution became 40% coarser, it could be more appropriate to use a larger (e.g., 4 X 3) array of 1 km-size MODIS pixels to cover a coarse-resolution SEVIRI pixel. My own guess is that a such modification would not bring substantial changes to the overall outcomes (e.g., it would not change which method is deemed best), and I am not certain that considering the exact pixel sizes and using 4X3 arrays of MODIS pixels would yield more appropriate comparisons to 3X3 arrays of SEVIRI HRV pixels. Even so, it seems important to clarify in the manuscript the actual SEVIRI resolution around Germany, and to discuss any limitations or problems the different pixel sizes may introduce into the comparison of small-scale variability in SEVIRI and MODIS data.

Additional suggestions for minor revisions are listed below:

Page 3, Line 4: The resolution of 2.1 $\mu$m MODIS data is 500 m (and not 1 km).

Page 5, Line 23: It could help to clarify that the subscripts 06, 08, and 16 indicate 0.6 $\mu$m, 0.8$\mu$m, and 1.6 $\mu$m.

Page 6, Lines 11-12: I suggest starting the paragraph with something like "As is it

discussed in Section 4,", just so readers know they will be able to learn about the exact estimation methods later on.

Page 6, Line 14: For added clarity, I suggest inserting "latter" in front of "variables".

Page 8, Lines 5-10: It would be interesting to add a few words about what may cause the variations in c. For example, could it be variations in typical cloud droplet size?

Page 14, Line 22: Wouldn't spatial averaging of MODIS data provide a better comparison than subsampling?

Page 14, Line 29: The part "(a)" seems to be missing from "Figure 8(a)".

Page 14, Line 31: The "t" in "table 1" should be capitalized.

Page 15, Line 4: The "s" in "section" should be capitalized.

Page 20, Line 9: It would help to clarify what is meant by SEVIRI LUT (what specific look-up table is referred to).

Page 20, Line 17: Some readers jump from the Abstract straight to the conclusions and read the rest only afterwards. For the sake of these readers, it is important to clarify in the conclusions section what is meant by the caret accent over tau and reff.

Page 20, Line 25: It would help to clarify that "local slopes" refer not to the slopes of the cloud top surface, but to the steepness of curves in the used LUT.

Page 21, Line 6: The spatial averaging used by MODIS is a reasonable alternative to downscaling. Although at visible wavelengths MODIS reflectances are available at a higher resolution, the MODIS cloud algorithm degrades the resolution of all input reflectances to a common 1 km resolution. Therefore, while downscaling could certainly help, the resolution mismatch can also be avoided by averaging, without the downscaling approach. Accordingly, at least the word "should" should be replaced.

Page 32, Lines 4-5 of Figure 2 caption: It would help to clarify that the blue lines

show the relative difference between the Constant Reflectance Ratio Approach and the resampled original data. To this end, the words "relative difference" should be included, and the mention of color should be moved to the end of the sentence.

---

## Author Comment (AC1) · 23 Dec 2019

We'd like to thank the editor for handling our manuscript, as well as reviewer #1 for reading our manuscript and providing numerous, helpful comments. We have carefully read through all the comments and questions and revised the manuscript accordingly. Please find our point-to-point response to reviewer #1 below. Here, the reviewer's general remarks are formatted to be left-aligned text in italic font, the specific questions/comments are shown in left-aligned text in bold and italic font, while our responses are indented and formatted in regular font.

Here is a summary of the major changes in the revised manuscript:

1) We rewrote the abstract to better summarize the results for $r_{\text{eff}}$, $W_L$ and $N_D$.
2) We rewrote the introduction so the connection to the study by *Deneke and Roebeling* (2010) becomes clearer.
3) We added a paragraph about the difference between spatial and optical resolution and made clear that we account for this difference by means of the modulation transfer function.
4) We removed Figures 4 and 6 and the respective text describing it.
5) In Section 6 we focus on 3 downscaling schemes (instead of 5) and simplified the experiment designations from "1a, 1b, 2a, 2b, 3a, 3b, 3c, and 3d" to the simpler naming scheme of "1, 2, and 3".
6) In section 6 we removed some of the statistical measures (i.e., the percentiles of retrieval differences) to simplify the analysis and focus on 4 statistical measures only.
7) We found a small bug in Figures 10 and 12 (now 8 and 10), where the nRD was normalized twice (and the factor 100 for the calculation of percentages) was applied twice. Naturally, this only affects the values, but not the interpretation.
8) We moved the VNIR-only versus full downscaling approach to its own Section.
9) The values for Table 2 also slightly changed (as for point 7, this did not affect the interpretation of results), as there was an additional filter applied that was not needed.
10) We rewrote parts of the conclusions and added more interpretation instead of just summarizing the findings of Section 6.

*Review of "Increasing the spatial resolution of cloud property retrievals from Meteosat SEVIRI by use of its high–resolution visible channel: Evaluation of candidate ap- proaches with MODIS observations" by Werner and Deneke.*

*The manuscript discusses a topic relevant for the scientific community. Various down- scaling techniques are presented and analysed in order to derive high resolution (1km) cloud properties from low resolution (3km) Meteosat SEVIRI data. Methods and motivations are layed out in detail, but the presentation suffers from the dizzying number of approaches used for downscaling, used for verification, of data spatial resolutions and of acronyms used. While I have full confidence in the authors scientific rigour, I was close to giving up reading through all the details offered. At the same time, I'm missing general interpretation of the large variety of results in some places. This leads to the unsatisfying point that the support of final and most important conclusions is not easy to find for the reader. I have the impression that this manuscript could be much improved by a major revision and tightening of the presentation, especially of the comparison results section 6.*

***Major points:***

***Especially section 6 is confusing. I would suggest to reduce the number of downscaling code versions. Especially the results section 6 even has versions not discussed anywhere. I would also suggest to reduce the number of error quantities discussed, maybe to the set shown in the tables Fig 10 and 12. Do not discuss other additional numbers in the text. Please see details below.***

We agree with the reviewer and changed the revised manuscript in the following ways:

1) We focused on 3 downscaling schemes: *Statistical Downscaling Approach, Constant Reflectance Ratio Approach*, and *Adjusted LUT approach with LUT Slope Adjustment*

2) We simplified the naming scheme of each experiment. Instead of 1b, 2b, 3b, 3c, and 3d we simply refer to the three downscaling schemes as 1, 2, and 3.

3) We only briefly summarize the performance of experiments 3b and 3c (i.e., the standard *LUT Approach* and the *Adjusted LUT* approach with *Adiabatic Assumptions*). No new experiment numbers are added for those and the summarized results are only a couple of sentences long.

4) We removed all mentions of the 1[st], 50[th], and 99[th] percentiles of the retrieval differences for each of the cloud variables. We likewise changed the correlation coefficient $R$ to explained variance $R^2$ in the scatterplots in Section 6, because this variable is used in the statistical comparison in the other two figures.

5) We moved the VNIR-only versus full downscaling part to its own section (Section 7). This (i) makes Section 6 shorter and easier to follow, and (ii) creates a more logical structure. After all, that new section does not compare different downscaling algorithms and instead looks at the effects of a scale-mismatch.

6) In the new Section 7 we removed the a and b experiment distinctions. Instead, we simply write out the respective experiments.

These steps should improve the readability of the comparison section. However, note that in the new Section 7 we kept the statistical measures (i.e., percentiles). We prefer the style of presentation by means of PDFs of the retrieval difference here and we believe that the statistical comparison by means of percentiles is appropriate.

***Is Deneke and Roebeling 2010 the basis of this paper? I had the impression down to page 8 that many things come from this older publication. If this is the case, it would be one possibility for shortening. You have to make the connection of the two clearer in the introduction.***

This study by *Deneke and Roebeling* (2010) introduced the statistical downscaling of VNIR reflectances by means of the HRV channel. It neither provided suggestions for downscaling of SWIR reflectances, nor did it analyze the effects on the subsequent cloud property retrievals.

Our study provides different approaches to downscale SWIR reflectances and evaluates the reliability of the resulting retrieval products. Moreover, it tests other downscaling techniques that ultimately yield improved results.

In this sense the *Deneke and Roebeling* (2010) study is more of a 'conversation starter'. It provides the basis of half of the *Statistical Downscaling* Approach (and parts of the LUT-based approaches).

We carefully rewrote that part of the introduction, which now says:
"The aim of this manuscript is to critically evaluate several candidate approaches for downscaling of the SEVIRI narrow–band reflectances for operational usage and to identify the most promising of these schemes, exploiting the fact that information on small-scale variability is available from its broadband high–resolution visible (HRV) channel. The study by *Deneke and Roebeling* (2010) presented a statistical downscaling approach of the SEVIRI channels in the visible to near-infrared (VNIR) spectral wavelength range. This method makes use of the fact that SEVIRI's high–resolution channel can be modelled by a linear combination of the 0.6 μm and 0.8 μm channels with good accuracy (*Cros et al.*, 2006). This study advances these efforts in three ways: (i) it explores other possible downscaling approaches, which might improve upon the statistical downscaling scheme, (ii) it introduces techniques to accurately capture information on the small–scale reflectance variability in the 1.6 μm–channel, which predominantly arises from variations in effective droplet radius, and (iii) it studies the impact of the various downscaling techniques on the subsequently retrieved cloud properties."

***Technical problem is that a companion paper (Deneke et al 2019) is obviosuly not submitted at this stage. I would recommend removal of all references to it or waiting for its puplication in discussion stage.***

We moved all references to the companion paper, which will be submitted at the end of January to the conclusion/outlook section. There, it basically works as a

'plans for future work' style reference. Since not a single part of our analysis depends on this (as of now) unpublished work, we think it is ok to mention it at least in the conclusions/outlook.

***Specific points***

***p.1, l. 11 ff: Where do these numbers for tau, reff and WL and ND come from? I can not easily find these numbers in the manuscript and I hardly can find any discussion of them. Please extend discussion of these later on or remove them from the abstract.***

Following other comments in this review, we removed the discussion of percentiles in section 6. This removes some of the clutter in the discussion and reduces the number of discussed statistics. However, we feel mentioning explained variance and interquartile ranges in the abstract (and to a lesser degree in the conclusions) is not really helpful. We believe the reader generally wants to know how big biases due to the lower spatial resolution are, in order to assess whether this is even an issue. We therefore prefer to keep these numbers in the abstract. We summarize these retrieval differences in the conclusions section: "The retrievals based on native–resolution reflectances (at a scale of $\approx 3$ km) are characterized by significant deviations from the reference retrievals, especially for $\hat{\tau}$ and $\hat{W}_L$. Here, random absolute deviations as large as $\approx 14$ and $\approx 89$ g m$^{-2}$ are observed, respectively (determined from the 1$^{st}$ or 99$^{th}$ percentiles of the absolute deviations between native and reference results for each cloud scene). For $\hat{r}_{eff}$ and $\hat{N}_D$ deviations of up to $\approx 6$ μm and $\approx 177$ cm$^{-3}$ exist, respectively."

***p.1, l. 20: The whole abstract reads as if it does not work very well. Maybe apart from tau. This concluding sentence reverses these statements. Please revise.***

We were overly cautious when describing the performance of the higher—resolution $r_{eff}$ retrieval in the abstract. We tried to convey the fact that the biases in the LRES retrieval are smaller than for $\tau$ and that the performance of the best downscaling technique is close to the baseline approach.

We agree with the reviewer that this is not necessary; after all there are still improvements for $r_{eff}$ and $N$, just not as prominent as those for $\tau$ and $W$L.

We revised parts of the abstract as follows:
"… Uncertainties in retrieved $r_{eff}$ at the native SEVIRI resolution are smaller and the improvements from downscaling the observations are less obvious than for $\tau$. Nonetheless, the right choice of downscaling scheme yields noticeable improvements in the retrieved $r_{eff}$. Furthermore, the improved reliability in retrieved cloud products results in significantly reduced uncertainties in derived $W_L$ and $N_D$. In particular, one downscaling approach provides clear improvements for all cloud products compared to those obtained from SEVIRI's standard-resolution and is recommended for future downscaling endeavors. This work advances efforts to mitigate impacts of scale mismatches among channels of multi–resolution instruments on cloud retrievals. "

***p. 3, l. 11: Isn't a clear reference to Deneke and Roebeling 2010 missing here? Can you please make that clear in the introduction.***

> As mentioned in the reply to a previous comment in this review, we rewrote that part of the discussion and clearly pointed out the connection to the study by *Deneke and Roebeling* (2010).

***p. 3, l. 20: This way you will only get relative errors. All problems retrievals at 1km resolution still suffer from are not discussed or improved . . . e.g. Zhang et al, 2009, 2011. Can you please mention that.***

> We agree with the reviewer. Indeed, after downscaling the reflectances (even if the results would be perfect), the same limitations exist that impact other 1 km–retrievals. In other words, the results likely would not represent the true cloud properties.

> We already discussed the effects of resolved and unresolved cloud variability and cited the respective literature. In the revised manuscript we added the following text to clarify this point:
> "Note, that even the retrieved cloud products from a hypothetically perfect downscaling technique would still be impacted by the effects of resolved and unresolved cloud variability. Therefore, the results of this study will not help to mitigate the uncertainties associated with the retrieval schemes of similar ≈ 1 km–sensors (e.g., clear–sky contamination, plane–parallel albedo bias, 3–dimensional radiative effects). "

***p. 3, introduction in general: Can you please make clearer: What is the motivation for an improved resolution of products? What problem do you expect to improve?***

> The manuscript states in the introduction that:
> "Use of the independent pixel approximation (IPA, see *Cahalan et al.*, 1994a, b) produces uncertainties in the retrieved cloud variables that are dependent upon the horizontal resolution of the observing sensor. "
> It subsequently describes the effects of resolved and unresolved cloud variability and cites the respective literature.

> Afterwards, it talks about the benefits of SEVIRI-like observations, as well as the disadvantage of the lower spatial resolution:
> "However, SEVIRI pixels are characterized by a lower spatial resolution of its narrow–band channels compared to other operational remote sensing instrumentation, like the Moderate Resolution Imaging Spectroradiometer (MODIS, *Platnick et al.*, 2003) or the Visible Infrared Imaging Radiometer Suite (VIIRS, *Lee et al.*, 2006). Given the increase in retrieval uncertainty due to the IPA constraints, there is a desire to increase the resolution for geostationary cloud observations. "

> We believe that this clearly motivates the study in our paper.

*p. 4, l. 30: "horizontal resolution of 3 x 3". This is the nominal sub-satellite resolution. Can you please make clear whether you consider the full spatial response function for each point. Much later it sounds like, but here you widely stick to the simplified "1 km", "3 km" without further explanation.*

Thanks for this comment; this point was similarly addressed by reviewer #2. The actual spatial resolution is dependent on the viewing geometry and thus on geolocation. By sticking with the simplified description of 3x3 km$^2$ and 1x1 km$^2$ we tried to make the manuscript less confusing, but apparently achieved the opposite.

Statistics of pixel size for the Germany domain are shown in Figure 1 of this reply. The 3x3 km$^2$ pixels are closer to 6.2x3.2 km$^2$, while the higher—resolution pixels cover an average area of 2.1x1.1 km$^2$. However, the factor 3 between the spatial resolutions of channels 1-3 and the HRV channel remain. Similar stretching is observed for the MODIS pixels of the four example scenes. For scene 1 pixels are 1.5x2.4 km$^2$ large (comparable to the SEVIRI HRV resolution), while the other scenes are characterized by 1.1x1.2 km$^2$ pixels.

[Figure]

*Fig 1: Statistics of SEVIRI pixel dimensions (in both latitude and longitude direction; i.e., South-North and East-West) for the native and HRV resolutions.*

We decided on a number of changes for the revised manuscript.
- We added the "≈" Symbol to the pixel scales in the abstract.

- We added "at the sub-satellite point and increases with higher sensor zenith angles" at the SEVIRI instrument description.
- We added a paragraph to the domain description in Section 2.3: "Due to the increased sensor zenith angles the spatial resolution of each SEVIRI pixel is degraded. The average pixel size is $6.20 \times 3.22$ km$^2$ and $2.06 \times 1.07$ km$^2$ for channels 1–3 and the HRV channel, respectively. To avoid confusion, we will use the designations LRES (abbreviation for lower–resolution) and HRES (abbreviation for higher–resolution) scales to refer to the $3\times3$km$^2$ and $1\times1$km$^2$ pixel resolutions from here on. "
- We replaced all other mentions of $1\times1$ km$^2$ and $3\times3$ km$^2$ with LRES and HRES abbreviations, or descriptive explanations. This should help avoid possible confusions by the reader.

With regard to the spatial response function, we added a paragraph in the SEVIRI instrument description. Here, we mention the difference between spatial and optical resolution for SEVIRI, the definition of the spatial response function, as well as the exact treatment in our study by means of the modulation transfer function. The latter describes the convolution of the sensor signal and the point spread function (i.e., the terms in the integral of the spatial response function) in Fourier space. See also our response to a latter comment in this reply for more details.

The added section says:
"As context for the present study, the reader is reminded that the spatial resolution of geostationary satellites is significantly reduced at higher latitudes due to the oblique viewing geometry. For Germany and Central Europe as considered in this paper, the pixel size is effectively increased by a factor of two in North–South direction as a result. In addition, the distinction between sampling and optical resolution needs to be acknowledged. While the former determines the distance between recorded samples, the latter is given by the effective area of the optical system, which is larger by a factor of 1.6 than the sampling resolution for SEVIRI (*Schmetz et al.,* 2002). The spatial response of optical systems is commonly characterized by their modulation transfer function, which describes the response of the optical system in the frequency domain.

Further information about the spectral width of each SEVIRI channel, as well as the respective spatial response and modulation transfer functions, can be found in *Deneke and Roebeling* (2010). "

***p. 5, l. 14/15 and 17: I do not understand the need for theses statements. These are purely technical, aren't they? First you mention a method not used in this study?! With modifications described in a study not published yet?! Then you are talking about a version control system development branch (?) to make clear that this algorithm is not perfectly the same as in the unpublished companion paper!? I doubt that the reader needs these documentation details.***

We agree with the reviewer's comment and removed the respective parts from the revised manuscript.

***p. 5, l. 28: Again the reader wants to know whether you consider the different spatial response function of normal and high resolution channels? Please comment why you think you do not need to consider this or how it is considered.***

The modulation transfer function, which describes the spatial response of an optical system in Fourier space, is applied during trigonometric interpolation. While the details are given in *Deneke and Roebeling* (2010), we added a reminder in the revised manuscript to ensure the reader that the spatial response of the sensor is considered:

"… implemented based on the discrete Fourier transform and multiplication with the modulation transfer function (see *Deneke and Roebeling*, 2010, for details) …"

***p. 6, l. 16: "Statistical downscaling": I do not see the "statistical" element? Are not all downscaling steps fully deterministic? No random element is in there? Please clarify.***

There are multiple techniques to downscale a coarse-resolution signal to a higher—resolution one. The simplest is an interpolation, which doesn't add any high-frequency variability to the new image. Another simple method would be the "Constant Reflectance Ratio Approach" discussed in this manuscript, which assumes a constant relationship between VNIR and SWIR reflectances for different spatial scales. From climate science we know of two more approaches:

(i)     Dynamical downscaling, which uses a higher-resolution model to estimate the smaller-scale information, and

(ii)     Statistical downscaling, which establishes statistical relationships between different data sets and uses the results in a multiple linear regression to predict the higher-resolution behavior.

We simply reminded the reader that the first technique in our manuscript is a *Statistical Downscaling Approach*, well known and used in other scientific fields. Please note that Technique #2 in our manuscript does not use any large-scale statistical relationships. Also, Technique #3 is a hybrid technique. While it uses statistical relationships between HRV and standard channels in the VNIR, it incorporates the shape of the SEVIRI LUT to determine the higher-resolution SWIR signal.

We believe that the later statement in the manuscript is sufficient to explain the naming scheme: "Note, that the use of linear models and bivariate statistics means that the downscaling algorithm described in this section is an example of statistical downscaling techniques, which are common in climate science applications (e.g., *Benestad*, 2011). "

*p. 7, l. 2: IQR=0.03 of what? Daily values? Hourly?*

All data points are included in this statistic, i.e., all 16-day intervals and each hourly data point. We updated the sentence in the revised manuscript as follows: "Considering all hourly data and each 16-day interval, the median …"

*p. 7, l. 4, Fig 2: I can only see three colors for 14:00, 15:00 and 16:00 UTC?!*

There is an overlap of data with values of about 0.51. For the 8-9 UTC and 13-16 UTC time stamps. The afternoon hours are just plotted on top of the morning hours. However, the old version of the manuscript said 15-16 UTC. We corrected this to 13-16 UTC. We also double checked all median and IQR values in this section (everything is correct here).

*p. 8, l. 7: "Diurnally, the variability in the hourly derived ..." : You mean the IQR is derived over 18 or 19 hourly values over one day? Or over 16 days?*

Thanks for pointing this sentence out; it indeed is confusing. We meant to say that during each 16-day interval, there is a huge diurnal variability in coefficient $c$, something not observed for coefficients $a$ and $b$. We changed that sentence to: "For each 16-day interval the variability in the hourly derived $c$ values…"

*p. 10, eq. 9: Under which assumption does eq. line 2 follow from eq. line 1?*

The long version of the equation is:

$$\frac{\hat{r}_{06,i} - r_{06}}{r_{06}} = \frac{\hat{r}_{HV,i} - r_{HV}}{r_{HV}}$$

$$\frac{\hat{r}_{06,i}}{r_{06}} - 1 = \frac{\hat{r}_{HV,i}}{r_{HV}} - 1$$

$$\hat{r}_{06,i} = \hat{r}_{HV,i} \cdot \frac{r_{06}}{r_{HV}}$$

The VNIR high-resolution signal is scaled by the ratio of SWIR to VNIR reflectance at the lower horizontal scale.

*p. 10, l. 15, "R ~ 1.0" : This proves that both approaches are rather equivalent, but suffer from the same core problem. the reff impact. Is this an important comparison or just a distracting sideshow?*

Given that Downscaling Techniques #1 and #2 are completely different approaches (statistical downscaling versus assuming a constant ratio for each individual pixel) it is actually quite remarkable that the results agree so well.

In an older version of the manuscript we put more emphasis on the actual reflectances. However, the main point of the manuscript is the impact on retrieval products, which naturally include the changes in the reflectances. We decided to remove that part in the revised manuscript, which shortens and hopefully improves the readability of the study.

***p. 10, l. 22: I can see that the first approach could produce negatives, but the second?***

Thanks for noticing that mistake. Indeed, technique 2 cannot produce negative values. This was a mistake of the simple counting algorithm we used, which treated a 0-length array as 1. The result for experiment 1 was not impacted. However, following the earlier comment we decided to remove that part from the revised manuscript (including the figure).

***p. 13, l. 1: Is this section including Fig 6 really needed? It confirms stuff that could be seen before and adds another side aspect.***

As mentioned earlier, in a prior version of the manuscript we put more emphasis on the actual reflectances, before comparing the impact on retrieval results. In order to shorten and streamline the manuscript, we agree that this section and the figure are not needed. We removed them in the revised manuscript.

***p. 14, l. 11: You did never mention failed retrievals and reasons for it. Skip this sentence?***

We agree with the reviewer and skipped this sentence in the revised manuscript.

***p. 14, l. 12, chapter: This section is confusing. I started reading with the understanding that you only use MODIS data in this chapter until I read the Fig. 8 caption which sounds like it shows SEVIRI data. Please make sure that this stays clear from the beginning and throughout the section. Do you use "SEVIRI data" or only "SEVIRI- resolution MODIS data" in this section?***

Thanks for noticing this error. The reviewer is correct in assuming that the evaluation of downscaling techniques is done exclusively with MODIS data, resampled on the SEVIRI HRV-channel grid. In an older version of this Figure, the RGB was constructed from downscaled SEVIRI reflectances, but this turned out to be confusing (because the analysis is done without SEVIRI data). Therefore, we changed the figure and constructed the RGB with resampled ~1-km MODIS data. Unfortunately, we did not change the caption of the figure. This has been corrected in the revised manuscript and that part of the caption now says: "RGB composite image of remapped MODIS channel 6, 2, and 1 reflectances at the horizontal resolution of SEVIRI's HRV channel at a horizontal scale of $1 \times 1$ $km^2$ at the sub–satellite point. Data is from example scene 1 sampled on 1 June 2013 at 10:05 UTC. "

***p. 14, l. 18: You mention spatial response for the first time here I guess. What about spectral differences between MODIS and SEVIRI? Please discuss.***

As mentioned in the response to an earlier comment, we added information about the modulation transfer function, which described the point spread function in Fourier space. The exact spatial response for each SEVIRI channel is considered throughout the manuscript.

Regarding the spectral differences between MODIS and SEVIRI: These differences do not affect our present study. We are not comparing operational MODIS C6 results to SEVIRI retrievals after downscaling. This technical study is purely performed with re-mapped MODIS data, as we are only interested in

evaluating the different downscaling techniques. In order to evaluate the different approaches, we require a common retrieval algorithm and a ground truth, which is provided by MODIS data.

We believe that this paragraph in the manuscript is sufficient to establish that goal: "It should be noted that retrievals based upon these radiances will be different than those based upon the original MODIS C6 radiances, or from an absolutely accurate representation of the (hypothetical) truly observed, high–resolution SEVIRI samples. For one, it uses the linear model of Cros et al. (2006) and Deneke and Roebeling (2010) as a proxy for the HRV channel, thereby excluding a potentially significant source of uncertainty. Moreover, MODIS acquires these reflectances under different viewing geometries (note that the true viewing angles are used in the CPP retrieval, so within the limits of plane–parallel radiative transfer, this effect is accounted for), and the spectral characteristics of the MODIS and SEVIRI channels are not entirely comparable. However, the goal of this study is to provide a consistent reference data set for a comparison of different retrieval data sets, which are derived from a single retrieval algorithm core. The actual absolute values of the retrieved cloud products are not important here. "

The companion paper, which will be submitted at the end of January 2020, will compare downscaled SEVIRI with operational MODIS C6 retrieval results (the statistical comparison for different cloud scenes shows a significant improvement between MODIS and SEVIRI due to the downscaling efforts). The applied downscaling theme was chosen based on the results of this study. This upcoming manuscript also presents other applications of the higher-resolution SEVIRI cloud products.

***p. 14, l. 33, "interpolation of SEVIRI samples": Are you talking about SEVIRI or MODIS data here? See point above.***

Again, thanks for noticing these inconsistencies. Again, the analysis in section 6 is performed exclusively with remapped MODIS data. No SEVIRI reflectances are included.

We carefully read through section 6 again and removed all ambiguities, to make sure that the reader knows that only MODIS data is considered in the analysis.

***p. 15, l. 4-8: Phew! Now you add sub-experiments "a, b, c, d" on top of the new nomenclature "1, 2, 3 " ... I'm struggling, to keep reading ... At least, do not use "1,2,3" acronyms alone, but write out the experiments to make them more recognizable. Please do not introduce experiments you will not even discuss (1a, 2a).***

We agree that the different nomenclatures are potentially confusing. We reduced the number of experiments to three (focusing on one of the options for the *LUT Approach*). We also removed versions a and b, which were discussed in Section 6.4 (now Section 7). When we discuss the difference between full downscaling and VNIR-only results, we write out the experiment description instead.

***p. 15, l. 11-12: And now ... a few new products on top. You have to mention the relevanve of these right in the introduciton.***

> We added the following information to the introduction, right after discussing resolved and unresolved variability in $\tau$ and $r_{eff}$: "These uncertainties are propagated to the liquid water content ($W_L$) and the droplet number concentration ($N_D$), which can be estimated from retrieved $\tau$ and $r_{eff}$. Estimates of $N_D$ are especially susceptible to uncertainties in $r_{eff}$, which impacts the reliability of aerosol–cloud–interaction studies (*Grosvenor et al.*, 2018). "

> Introducing these variables early should help the reader understand their importance and the need to include these parameters in the downscaling analysis.

***p. 15, l. 21: This is not the first time you use the exact spatial response functions, isn't it? This is rather late to mention the reference for the first time.***

> As mentioned earlier, we added a paragraph about the modulation transfer function and its relationship to the point spread function and spatial response in the SEVIRI-section. In the revised manuscript we also use the modulation transfer function throughout the manuscript, which should help avoid confusion.

***p. 15, l. 23, "3x3 block". This block is 333 m resolution here, right? Please make sure that this can not be confused with the other 3x3 blocks mentioned before.***

> That is correct. We added the following in parentheses in the revised manuscript: "(each pixel with a horizontal resolution of 333 m)"

***p. 15, l. 25, "level 1b": Could be easily confused with your experiment notation. You did not use the term level 1b data before, you do not need it here.***

> As mentioned earlier, we removed the a, b, c, d, subcategories and focus on 3 experiments on the revised manuscript. The MODIS level 1b radiances are first mentioned at the start of Section 6.1:
> "To obtain a reliable higher–resolution reference data set, MODIS level 1b swath observations (MOD021km) have been projected to the grid of the SEVIRI HRV reflectance observations …".
> Since this is the correct reference to the data set, together with the easier experiment description, we believe that any potential confusion is now avoided.

***p. 15, l. 28-29: What is the "modulation transfer function" good for? Why do you only mention it here, that late in the manuscript?***

> Thanks for this comment. Without proper context, this was indeed confusing. As mentioned earlier, we added a section about the modulation transfer function, in the SEVIRI-section.

> The discussion in *Deneke and Roebeling* (2010) points out that the optical resolution of the SEVIRI channels is lower than the spatial resolution by a factor of about 1.6. This means that the signal for each pixel is not only determined by the observations within the nominal sampling resolution (i.e., the pixel itself), but

also includes contributions from neighboring pixels. This characteristic is effectively described by the spatial response function $S$ of the respective SEVIRI channel:

$$S(x_0) = \int_A w(x - x_0)L(x)dx$$

$$S(x_0) = (w * L)(x_0)$$

where $x_0$ is the displacement from the center of the field of view, $L(x)$ is the radiance at position x, $w$ is a weighting function commonly referred to as the point spread function, and $*$ indicates the convolution of $w$ and $L$. Applying the Fourier convolution theorem means that this convolution is equivalent to a multiplication of the Fourier transforms of L and w. The modulation transfer function, which fully describes the spatial response of an imager, is the modulus of the Fourier transform of w.

As mentioned in an earlier reply, we added a new paragraph to the revised manuscript, which explains the treatment of optical resolution.

***p. 16, l. 2, "spectral characteristics": This is again too late to mention such an obvious problem that late.***

This might be the result of some of the confusion regarding the use of MODIS and SEVIRI data. The evaluation of downscaling techniques is based exclusively on MODIS data, which is available at ~1 km resolution. We remap this dataset to the SEVIRI geometry and apply the SEVIRI retrieval code to it. We do not expect these results to agree with operational MODIS C6 products, as the use of a different algorithm core alone will yield different results.

However, this is not the focus of this manuscript, as we are not interested in absolute values of the retrieval results. We are only interested in a comparison to the reference results. We mention this in the manuscript, when we say: "However, the goal of this study is to provide a consistent reference data set for a comparison of different retrieval data sets, which are derived from a single retrieval algorithm core. The actual absolute values of the retrieved cloud products are not important here. "

A comparison between downscaled SEVIRI retrievals, employing the most promising technique revealed by this study, and operational MODIS C6 results is performed in the follow-up paper (amongst demonstrations of other applications of the new dataset).

***p. 17, l16ff: I'm missing this kind of more conclusive interpretation elements instead of adding number to number in the text.***

As mentioned earlier, we removed all mentions of the 1st, 50th and 99th percentiles in Sections 6.2 and 6.3. We added some additional interpretation of the reason

behind the shortcomings of the different downscaling techniques both in Section 6.2 and 6.3, as well as in the conclusions (see our replies to later comments below).

***p. 17, l.31: This sounds as if 2/3 and 5/9 are magic numbers found empirically. Approx. 2/3 follow from optical properties directly and in general. 5/9 contains an empirical element. I would prefer to say "WL=2/3 ro tau reff" and adiabatic clouds have a typical additional factor of factor=5/6 due to their vertical structure.***

We changed the manuscript as follows:

"$W_L \approx \frac{2}{3} \cdot \rho_L \cdot \tau \cdot r_{eff}$ .

Here, $\rho_L$ is the bulk density of liquid water. Assuming adiabatic clouds, where the vertical structure of effective droplet radius follows the adiabatic growth model, introduces an extra factor of 5/6 and the coefficient 2/3 changes to 5/6 · 2/3 = 5/9."

***p. 18, l. 3, eq. 18: Again, this equation seems to contain magic, but is rather simple in its core. Maybe give some additional explanation: "Droplet number could simply be derived from LWC and a droplet size. Using empirical factors accounting for typical droplet size distributions and vertical cloud structure, the following can be derived:"***

We slightly disagree with the reviewer in this point. The assumptions going into the derivation of droplet number concentration are not trivial and include terms for the condensation rate, shape of the droplet number size distribution and more. Assumptions about subadiabaticity alone change statistics of droplet number concentration substantially. Likewise, going into detail about the derivation does not improve the readability of the manuscript. We believe it is enough to cite the appropriate literature here.

However, we added some clarification about the nature of the assumptions and this part of the manuscript now reads:
"Calculating $N_D$ from remote sensing products requires a number of assumptions, e.g., about the vertical cloud structure and shape of the droplet number size distribution, which are summarized and discussed in *Brenguier et al.* (2000); *Schüller et al.* (2005); *Bennartz* (2007); *Grosvenor et al.* (2018). A simplified form of the resulting equation for $N_D$ is:"

***p. 18, l. 22, "3c overall performs worst": Why? Can you give a general explanation or guess?***

As mentioned earlier, we made several changes to this section. We focus on experiments 1b, 2b, and 3d (now just 1, 2, and 3) and only briefly summarize the results for 3b and 3c.

We added a general explanation for the poor performance of 3c (and 3b in comparison) at several points in the revised manuscript:

"... We believe that this might be caused by the sensitivity of the cloud property retrieval to small reflectance perturbations, in particular for broken clouds. It is also an indication that assuming constant subpixel $r_{eff}$ values within each LRES

pixel is not sufficient. We plan to investigate this effect further in future studies. However, the second *Adjusted LUT Approach*, which determines SWIR reflectance adjustments based on adiabatic theory, performs even worse ($R^2$ of 0.846, 0.579, 0.741, ad 0.519 for cloud scenes 1–4, respectively). This suggests that the observed cloud fields do not follow adiabatic theory and the method is not adequate to estimate higher–resolution $\hat{r}16$. "

And:

"As before, we also tested the standard *LUT Approach* highlighted in Section 4.3, as well as the second *Adjusted LUT Approach*, which determines SWIR reflectance adjustments based on adiabatic theory. Due to the poor performance of the $\hat{r}\text{eff}$ retrieval, the $\hat{N}D$ results based on adiabatic assumptions show a similarly poor agreement to the reference results. Meanwhile, the cloud variables based on the standard *LUT Approach* never show the best or worst performance, but are almost universally worse than the *Adjusted Lookup Table Approach* with *LUT Slope Adjustment*. This again illustrates that assumptions of adiabatic clouds and constant subpixel $r_{\text{eff}}$ values within each LRES pixel are not suitable for the cloud scenes analyzed in this study. "

The poor performance of adiabatic assumptions is not surprising. After all, the literature is filled with examples of remote sensing studies that show non-adiabatic behavior. Here is an example of MODIS data for example scene 1:

[Figure]

*Fig 2: Example of $\tau$-$r_{\text{eff}}$ relationships for example cloud scene 1.*

The grey dashed line indicates the adiabatic relationship reported by *Szczodrak et al.* (2001):

$$\ln(r_{\text{eff}}) = \alpha \log(\tau) + \beta$$

There clearly are a multitude of data points not following that relationship. Here are two more examples from ASTER observations over altocumulus and from *Suzuki et al.* (2006):

[Figure]

*Fig 3: Examples for τ-$r_{eff}$ relationships. (left panel) ASTER retrievals for a 50x50 km² scene. (right panel) From Suzuki et al. (2006).*

It is overall not surprising that adiabatic assumptions are not ideal to describe all the different cloud types observed in the different cloud scenes shown in the manuscript.

***p. 19, l. 25, "results provide strong evidence that simulateneous downscaling of the SWIR reflectances is essential": Again, why? Can you give a general explanation or guess?***

Not downscaling the SWIR reflectance basically means that VNIR and SWIR reflectances exist at different spatial scales. Figure 8 in Werner et al. (2018b) compares ASTER SWIR reflectances at 240 m to artificially degraded (to 960 m) ones, as well as to values from the *Constant Reflectance Ratio Approach* (see Figure 4 below in this reviewer reply). There are significant deviations between the 240 m and 960 m SWIR reflectance, while the *Constant Reflectance Ratio Approach* provides a good estimate of the true 240 m results with a significantly reduced normalized root mean square deviation (nRMSD in that plot).

Naturally, assuming a wrong SWIR reflectance has a substantial impact on the $r_{\text{eff}}$ retrieval, but even the cloud optical thickness will be impacted (because the

isolines are generally not perpendicular; see the example SEVIRI LUT in this manuscript). It is therefore understandable that a retrieval with a scale mismatch should be avoided.

In the revised manuscript we added the following information:

"This confirms the findings in Werner et al. (2018b), who illustrated that SWIR reflectances differ significantly between the pixel-level and subpixel scale and that reliable cloud property retrievals should avoid scale mismatches between the reflectances from the VNIR and SWIR channels."

[Figure]

*Fig 4: (a) Comparison between observed 240 m SWIR reflectances and 960 m observations, replicated to each 240 m subpixel. (b) Comparison between observed 240 m SWIR reflectances and the results of the Reflectance Ratio Approach. Adapted from Werner et al. (2018b).*

***p. 20, chapter: The conclusions section also needs more of this kind of general explanations and interpretations instead of repeating "x is better than y, so use x".***

We extensively rewrote the conclusions section, especially the summary of the downscaling performance. We shortened the summary of the results and added some general interpretation instead.

Some of these explanations are listed below:

"This improvement can be attributed to the use of higher–resolution reflectances, which resolve the large–scale variability of the scene. It is shown that either downscaling approach, which applies estimates of the unresolved small–scale variability to the reflectance field, yields reliable retrievals of $\hat{\tau}$ at the horizontal resolution of the SEVIRI HRV channel. "

And:

"The former technique relies on large–scale statistical relationships between the reflectances, which might vary with the size of the observed region, prevalence of different cloud types and viewing geometry. The latter technique, meanwhile, was

developed for optically thin clouds, where the relationship between VNIR and SWIR reflectance can be approximated by a linear function (Werner et al., 2018b). Conversely, for more homogeneous altocumulus fields the *LUT Approach* with *Adiabatic Adjustment* seems inadequate and yields the worst comparison to the reference effective radius. The study by *Miller et al.* (2016), following similar studies, illustrated that drizzle and cloud top entrainment yield vertical cloud profiles closer to homogeneous assumptions and away from the adiabatic cloud model. Similar processes might affect the retrieval for the presented cloud scenes in this study. "

And:

"Due to the fact that these variables are derived from retrieved $\hat{\tau}$ and $\hat{r}_{\text{eff}}$, a similar behavior is observed for the derived $\hat{W}_L$ and $\hat{N}_D$. "

***p. 21, l. 5/6: Many studies show that going below the 1 km scale might introduce new problems with variability which are smoothed out at this scale. Please discuss this caveat when making such a suggestion.***

This suggestion was indeed not well written.

First of all, the spatial mismatch we mentioned is a direct result of the downscaling approach, which is the focus of our study (i.e., the resolution mismatch did not exist before downscaling the VNIR reflectances, yet we discussed downscaled reflectances for the purpose of this study). However, MODIS, e.g., does not have a spatial mismatch, because the VNIR data is aggregated to the horizontal resolution of the SWIR signal. We simply meant to say that if downscaling is performed, it is essential to also downscale the SWIR band reflectance, not just the VNIR band observations.

The second issue is that downscaling and retrieving at the VNIR resolution might put us close to the radiative smoothing scale; below that scale (about 200-400m, according to *Davis et al.*, 1997) the reflected field is characterized by enhanced radiative smoothing and the retrievals might be impacted by 3D radiative effects. Naturally, these facts need to be considered, before a decision about downscaling is made.

We rewrote the respective paragraph and it now says:
"This illustrates that, in order to achieve reliable higher–resolution retrievals, all channels need to capture small–scale cloud heterogeneities at the same scale. These results confirm the findings of Werner et al. (2018b), who compared SWIR reflectances at different spatial scales and demonstrated the need for effective downscaling approaches to match the spatial scale of the VNIR reflectance. This also has implications for other multi–resolution sensors, such as MODIS, VIIRS, and GOES–R ABI. To avoid a scale–mismatch of resolved variability in the VNIR and SWIR channels, the higher–resolution observations can either be degraded to match the lower–resolution samples (which yields overall lower–

resolution cloud property retrievals), or downscaling techniques are applied to one or both channel reflectances, which yields matching scales and higher–resolution estimates of cloud properties. It is important to note that downscaling might result in increased retrieval uncertainties, if the spatial resolution is below the radiative smoothing scale ($\approx 200 - 400$ m, see *Davis et al.*, 1997)."

**Fig. 9, caption: Here "1b" is mentioned. Isn't it "2b" in the text?**
Thanks for noticing this mistake. It is indeed experiment 2b (just 2 in the revised manuscript). We corrected this mistake, which was also present in the caption of Figure 11.

**Fig 10.: Please write out the experiments in words in addition to number codes.**
We not only reduced the number of experiments shown (-2), but also included the experiment description in the caption of both Figure 10 and 12 (now 8 and 10 in the revised manuscript):
"… downscaling experiments 1 (statistical downscaling approach), 2 (*Constant Reflectance Ratio Approach*), and 3 (*Adjusted Lookup Table Approach with LUT Slope Adjustment*)…"

**Fig. 6 and 13.: Needed?**
We removed both Figure 4 and 6 in the revised manuscript. Both Figures indicated changes in the reflectance, but were not really needed for the retrieval comparison and conclusions. However, Figure 13 is needed to highlight the importance of simultaneous downscaling of the VNIR and SWIR reflectance.

**References**

Suzuki, K., Nakajima, T., Nakajima, T. Y., and Khain, A.: Correlation pattern between effective radius and optical thickness of water clouds simulated by a spectral bin micro- physics cloud model, SOLA, 2, 116–119, 2006.

Szczodrak, M., Austin, P., and Krummel, P.: Variability of optical depth and effective radius in marine stratocumulus clouds, J. Atmos. Sci., 58, 2912–2926, 2001.

Werner, F., Zhang, Z., Wind, G., Miller, D. J., Platnick, S., and Di Girolamo, L.: Improving cloud optical property retrievals for partly cloudy pixels using coincident higher-resolution single band measurements: A feasibility study using ASTER observations, J. Geophys. Res. Atmos., 123, 12,253–12,276, https://doi.org/10.1029/2018JD028902, https://agupubs.onlinelibrary.wiley.com/doi/abs/10.1029/2018JD028902, 2018b.

---

## Author Comment (AC2) · 23 Dec 2019

We'd like to thank the editor for handling our manuscript, as well as reviewer #2 for reading our manuscript and providing numerous, helpful comments. We have carefully read through all the comments and questions and revised the manuscript accordingly. Please find our point-to-point response to reviewer #2 below. Here, the reviewer's general remarks are formatted to be left-aligned text in italic font, the specific questions/comments are shown in left-aligned text in bold and italic font, while our responses are indented and formatted in regular font.

Here is a summary of the major changes in the revised manuscript:

1) We rewrote the abstract to better summarize the results for $r_{eff}$, $W_L$ and $N_D$.
2) We rewrote the introduction so the connection to the study by *Deneke and Roebeling* (2010) becomes clearer.
3) We added a paragraph about the difference between spatial and optical resolution and made clear that we account for this difference by means of the modulation transfer function.
4) We removed Figures 4 and 6 and the respective text describing it.
5) In Section 6 we focus on 3 downscaling schemes (instead of 5) and simplified the experiment designations from "1a, 1b, 2a, 2b, 3a, 3b, 3c, and 3d" to the simpler naming scheme of "1, 2, and 3".
6) In section 6 we removed some of the statistical measures (i.e., the percentiles of retrieval differences) to simplify the analysis and focus on 4 statistical measures only.
7) We found a small bug in Figures 10 and 12 (now 8 and 10), where the nRD was normalized twice (and the factor 100 for the calculation of percentages) was applied twice. Naturally, this only affects the values, but not the interpretation.
8) We moved the VNIR-only versus full downscaling approach to its own Section.
9) The values for Table 2 also slightly changed (as for point 7, this did not affect the interpretation of results), as there was an additional filter applied that was not needed.
10) We rewrote parts of the conclusions and added more interpretation instead of just summarizing the findings of Section 6.

*My opinion is that this manuscript presents significant work well worth publishing. The key achievement lies the development and testing of methods for using geostationary satellite data to obtain cloud properties at a three times higher spatial resolution than the current standard. The methodology is sound, and the presentation is generally clear. I recommend a number of minor refinements (mainly to improve clarity), but there is one issue I'd like to single out in particular.*

**The text says throughout the manuscript (starting with Lines 3-4 of the abstract) that the proposed methods can increase the spatial resolution of SEVIRI cloud products from 3 km to 1 km (from the resolution of most SEVIRI bands to the resolution of the SEVIRI HRV band). My understanding, however, is that the resolution of SEVIRI observations is 3 km and 1 km only at the sub-satellite point, and that this resolution degrades with the cosine of the viewing zenith angle. (See, for example, http://www.esa.int/esapub/bullet111/chapter4_bul111.pdf or http://www.icare.univ-lille1.fr/projects/seviri-aerosols.) For the test area around Germany, this can increase the meridional extent of SEVIRI pixels by 40% or more. For the most part, considering this effect would require only a clarification in the text; the only part where this becomes a substantial issue is the comparison with MODIS data. Considering that the meridional resolution of MODIS images should remain around 1 km even if the SEVIRI resolution became 40% coarser, it could be more appropriate to use a larger (e.g., 4 X 3) array of 1 km-size MODIS pixels to cover a coarse-resolution SEVIRI pixel. My own guess is that a such modification would not bring substantial changes to the overall outcomes (e.g., it would not change which method is deemed best), and I am not certain that considering the exact pixel sizes and using 4X3 arrays of MODIS pixels would yield more appropriate comparisons to 3X3 arrays of SEVIRI HRV pixels. Even so, it seems important to clarify in the manuscript the actual SEVIRI resolution around Germany, and to discuss any limitations or problems the different pixel sizes may introduce into the comparison of small-scale variability in SEVIRI and MODIS data.**

The reviewer is absolutely correct. The actual spatial resolution is dependent on the viewing geometry and thus on geolocation. By sticking with the simplified description of 3x3 $km^2$ and 1x1 $km^2$ we tried to make the manuscript less confusing, but apparently achieved the opposite.

Statistics of pixel size for the Germany domain are shown in Figure 1 of this reply. The 3x3 $km^2$ pixels are closer to 6.2x3.2 $km^2$, while the higher—resolution pixels cover an average area of 2.1x1.1 $km^2$. However, the factor 3 between the spatial resolutions of channels 1-3 and the HRV channel remain. Similar stretching is observed for the MODIS pixels of the four example scenes. For scene 1 pixels are 1.5x2.4 $km^2$ large (comparable to the SEVIRI HRV resolution), while the other scenes are characterized by 1.1x1.2 $km^2$ pixels.

[Figure]

*Fig 1: Statistics of SEVIRI pixel dimensions (in both latitude and longitude direction; i.e., south-north and east-west) for the native and HRV resolutions.*

With regard to the evaluation of the downscaling techniques, these differences have no effect. There are two reasons for that: (i) We do not aggregate/colocate the MODIS data on the SEVIRI geometry. Instead, we first interpolate the MODIS reflectances on a higher-resolution grid and subsequently re-map these higher-resolution samples with the help of the sensor characteristics and open-source gdal libraries. (ii) We do not compare SEVIRI to MODIS; in fact, the actual values of the re-mapped MODIS reflectances are not important. They simply serve as a ground-truth for SEVIRI $r_{06}$, $r_{08}$ and $r_{16}$ reflectances at the HRV geometry, which is subsequently degraded (using the SEVIRI spatial response characteristics) by means of the same Fourier transforms (i.e., trigonometric interpolation) we describe throughout the manuscript. In other words, we degrade a ground-truth according to the SEVIRI characteristics and subsequently try to replicate the ground-truth again by means of the different downscaling techniques.

An actual comparison between downscaled SEVIRI and operational MODIS results is presented in the companion paper, which will be submitted by the end of January 2019 (this paper will also present other applications for this high-res SEVIRI data set). Here, we are just interested in finding a suitable technique.

We decided on a number of changes for the revised manuscript.
- We added the "≈" Symbol to the pixel scales in the abstract.
- We added "at the sub-satellite point and increases with higher sensor zenith angles" at the SEVIRI instrument description.
- We added a paragraph to the domain description in Section 2.3: "Due to the increased sensor zenith angles the spatial resolution of each SEVIRI pixel is degraded. The average pixel size is $6.20 \times 3.22$ km$^2$ and $2.06 \times 1.07$ km$^2$ for channels 1–3 and the HRV channel, respectively. To avoid confusion, we will use the designations LRES (abbreviation for lower–resolution) and HRES (abbreviation for higher–resolution) scales to refer to the $3 \times 3$km$^2$ and $1 \times 1$km$^2$ pixel resolutions from here on. "
- We replaced all other mentions of $1 \times 1$ km$^2$ and $3 \times 3$ km$^2$ with LRES and HRES abbreviations, or descriptive explanations. This should help avoid possible confusions by the reader.

*Additional suggestions for minor revisions are listed below:*
*Page 3, Line 4: The resolution of 2.1 µm MODIS data is 500 m (and not 1 km).*

Thanks for noticing this mistake. We corrected that error and it now says:
"250 m horizontal resolution versus 500 m for the 0.6 µm and 2.1 µm channels, respectively"

*Page 5, Line 23: It could help to clarify that the subscripts 06, 08, and 16 indicate 0.6 µm, 0.8µm, and 1.6 µm.*

We actually mention that in the SEVIRI description in section 2.1, where it says:
"The two VNIR reflectances ($r_{06}$ and $r_{08}$) are sampled in bands 1 and 2, respectively, and are centered around wavelengths $\lambda = 0.635$µm and $\lambda = 0.810$µm. SWIR reflectances ($r_{16}$) are provided by channel 3 observations, which are centered around $\lambda = 1.640$ µm."

*Page 6, Lines 11-12: I suggest starting the paragraph with something like "As is it discussed in Section 4,", just so readers know they will be able to learn about the exact estimation methods later on.*

We added the following before that paragraph:
"As is discussed in sections 4.1–4.4, the derived reflectances…"

*Page 6, Line 14: For added clarity, I suggest inserting "latter" in front of "variables".*

We added the word "latter", as suggested.

*Page 8, Lines 5-10: It would be interesting to add a few words about what may cause the variations in c. For example, could it be variations in typical cloud droplet size?*

We agree with the reviewer that it is worthwhile to discuss the behavior of parameter c a bit more.

The answer can be found in the shape of the SEVIRI LUT (see Figure 2 of this response, which is adapted from the manuscript). For a constant effective radius and increasing VNIR reflectance ($r_{06}$), which indicates an increase in cloud optical thickness, the SWIR reflectances ($r_{16}$) at first increase almost linearly ($r_{06}$ < 0.3). However, for $r_{06} > 0.3$ there is a curvature in the isolines and the linear relationship between $r_{16}$ and $r_{06}$ becomes non-linear. For even larger optical thicknesses ($r_{06} > 0.7$) the isolines become orthogonal and $r_{16}$ remains constant with increasing $r_{06}$. For the latter case positive or negative changes in subpixel VNIR reflectances would be translated into positive and negative SWIR reflectance deviations, even though for large optical thicknesses $r_{16}$ becomes independent of $r_{06}$. This means that assuming a linear relationship in the form $\langle r_{HV} \rangle = c \cdot r_{16}$ is a flawed assumption outside of optically thin clouds.

Thus, scenes with convective clouds, where the optical thickness can be larger than 20 (even larger than 100) are not well described by this relationship. As a result, the fit coefficient $c$ is not well constrained and can vary widely from hour to hour. However, stratus and altocumulus cloud fields are usually characterized by $\tau \approx 10$ and for these types of clouds this relationship should work rather well. As a result, varying cloud types will determine the reliability of this relationship. Over central Europe we often observe altocumulus and stratus clouds and thus for a large number of pixels the linear relationship works quite well (see the dark red and silver area around the 1:1 line in Figure 3b of the manuscript). For small cumulus clouds and convective thunderstorms, however, we will get large deviations from the linear relationship.

[Figure]

*Fig 2: Example SEVIRI LUT. Isolines for constant $\tau$ and $r_{eff}$ are shown in dashed gray lines.*

In the revised manuscript we added the following explanation:

"This behavior is expected, as the relationship between VNIR and SWIR reflectance can usually not be described by a linear function (see discussions in *Werner et al.,* 2018a, b, as well as the LUT examples in Figure 4 later in this study). For a constant $r_{eff}$ there is a linear increase in $r_{16}$ with increasing $r_{06}$, as the cloud optical thickness increases. However, the slope of this linear relationship increases with decreasing $r_{eff}$. For $\tau > 10$ the relationship between $r_{16}$ and $r_{06}$ is characterized by a prominent curvature, while for $\tau \gg 10$ the $r_{16}$ become independent of $r_{06}$. Therefore, the fit coefficients $c$ depend on the distribution of cloud optical and microphysical parameters, which varies widely with cloud type, meteorological conditions and different dynamic processes.

***Page 14, Line 22: Wouldn't spatial averaging of MODIS data provide a better comparison than subsampling?***

Thanks for this comment. This part of the manuscript was actually a bit confusing in the original manuscript.

It turns out that trigonometric interpolation (i.e., Fourier transform of the image and the inverse on a higher-resolution grid), combined with the application of the modulation transfer function (i.e., the spatial response function in Fourier space) yields an interpolated image, where the reflectance of the central pixel of each 3x3 pixel block corresponds to the lower—resolution reflectance value. In other words, by subsampling we combine the effects of spatial and optical resolution of the SEVIRI imager and get the exact reflectances that the lower-resolution SEVIRI channels would see. By carefully applying the two different modulation transfer functions (from the HRV channel and channels 1-3) and subsampling of the central pixel of each 3x3 pixel block we could simulate the reflectances at the lower spatial resolution (i.e., the native resolution of SEVIRI channels 1-3).

However, this is not the pathway we chose for this study. As mentioned in Section 6.1, we generated a second data set, where the MODIS level 1b observations where remapped to the lower-resolution (~3 km) grid (in the same way the reference data set was created at the HRV grid). The baseline results where then calculated by trigonometric interpolation and smoothing with the modulation transfer function.

Note, that both pathways are valid and yield the same baseline reflectances.

Somehow, the old manuscript version described both pathways and the result was rather confusing. We rewrote parts of both the general introduction to Section 6, as well as Section 6.1, where the remapping is described:

"Remapping MODIS reflectances to SEVIRI's LRES grid (i.e., the native resolution of channels 1–3) subsequently provides the means to apply the various downscaling schemes, as well as the simple triangular interpolation approach, in order to compare the retrieved cloud products (i.e., $\hat{\tau}$ and $\hat{r}_{eff}$, as well as $\tilde{\tau}$ and $\tilde{r}_{eff}$) to the reference results. Naturally, the ideal downscaling approach would

yield results that closely resemble the MODIS–provided HRES observations. Furthermore, the ideal downscaling approach would also represent an improvement upon the simple interpolation technique. The reader is reminded, that the latter data are still available at a higher resolution than the native LRES grid of the SEVIRI $r_{06}$, $r_{08}$, and $r_{16}$ channels, but no longer contain any information about the high–frequency reflectance variability. As the simplest approach to derive higher–resolution cloud products, these results are called the baseline results. "

And:

"To perform the subsequent downscaling experiments, a second set of level 1b radiances are generated, where the spatial variability is reduced to match that of the LRES–channels of Meteosat SEVIRI. This step again involves the smoothing of the respective reflectance field with the channel–specific modulation transfer function of the lower–resolution SEVIRI channels (EUMETSAT, 2006). This data set represents hypothetical SEVIRI–like observations at the native LRES resolution. "

***Page 14, Line 29: The part "(a)" seems to be missing from "Figure 8(a)".***
Thanks for pointing out this mistake, we added "(a)" to the text.

***Page 14, Line 31: The "t" in "table 1" should be capitalized.***
We capitalized the "t". We also capitalized it for other table references in the manuscript.

***Page 15, Line 4: The "s" in "section" should be capitalized.***
We capitalized the "s". We also capitalized it for other section references in the manuscript.

***Page 20, Line 9: It would help to clarify what is meant by SEVIRI LUT (what specific look-up table is referred to).***
Given the next comment (that a lot of readers jump from the Abstract to conclusions), we agree to add a clarification here. We added the following information in parentheses after "SEVIRI LUT":
***"***(which consists of simulated SEVIRI reflectances for different viewing geometries and combinations of cloud properties)".

***Page 20, Line 17: Some readers jump from the Abstract straight to the conclusions and read the rest only afterwards. For the sake of these readers, it is important to clarify in the conclusions section what is meant by the caret accent over tau and reff.***
Again, we agree. We added the following information in parentheses:
***"***(i.e., the actual higher-resolution cloud properties)"

***Page 20, Line 25: It would help to clarify that "local slopes" refer not to the slopes of the cloud top surface, but to the steepness of curves in the used LUT.***

We changed the sentence as follows:
"with an adjustment based on the calculation of isoline slopes in the SEVIRI LUT".

***Page 21, Line 6: The spatial averaging used by MODIS is a reasonable alternative to downscaling. Although at visible wavelengths MODIS reflectances are available at a higher resolution, the MODIS cloud algorithm degrades the resolution of all input reflectances to a common 1 km resolution. Therefore, while downscaling could certainly help, the resolution mismatch can also be avoided by averaging, without the downscaling approach. Accordingly, at least the word "should" should be replaced.***

We agree with the reviewer, even though the spatial mismatch is a direct result of the downscaling approach, which is the focus of this study (i.e., the resolution mismatch did not exist before downscaling the VNIR reflectances, yet we want downscaled reflectances for the purpose of this study). The sentence is indeed misleading. We meant to say that it is essential to also downscale the SWIR band reflectance, not just the VNIR band ones.

We rewrote the respective paragraph and it now says:
"This illustrates that, in order to achieve reliable higher–resolution retrievals, all channels need to capture small–scale cloud heterogeneities at the same scale. These results confirm the findings of Werner et al. (2018b), who compared SWIR reflectances at different spatial scales and demonstrated the need for effective downscaling approaches to match the spatial scale of the VNIR reflectance. This also has implications for other multi–resolution sensors, such as MODIS, VIIRS, and GOES–R ABI. To avoid a scale–mismatch of resolved variability in the VNIR and SWIR channels, the higher–resolution observations can either be degraded to match the lower–resolution samples (which yields overall lower–resolution cloud property retrievals), or downscaling techniques are applied to one or both channel reflectances, which yields matching scales and higher–resolution estimates of cloud properties. It is important to note that downscaling might result in increased retrieval uncertainties, if the spatial resolution is below the radiative smoothing scale ($\approx 200 - 400$ m, see *Davis et al.*, 1997)."

***Page 32, Lines 4-5 of Figure 2 caption: It would help to clarify that the blue lines show the relative difference between the Constant Reflectance Ratio Approach and the resampled original data. To this end, the words "relative difference" should be included, and the mention of color should be moved to the end of the sentence.***

Thanks for this comment. In the revised manuscript we decided to remove that figure and the respective section discussing it (following advice from reviewer #1).